# Dominant patterns of interaction between the tropics and mid-latitudes in boreal summer: Causal relationships and the role of timescales

Giorgia Di Capua[1,2], Jakob Runge[3], Reik V. Donner[1,4], Bart van den Hurk[2,5], Andrew G. Turner[6,7], Ramesh Vellore[8], Raghavan Krishnan[8], and Dim Coumou[1,2]

[1]Potsdam Institute for Climate Impact Research, Potsdam, Germany
[2]VU University of Amsterdam, Institute for Environmental Studies, Amsterdam, Netherlands
[3]German Aerospace Centre, Institute of Data Science, Jena, Germany
[4]Magdeburg-Stendal University of Applied Sciences, Magdeburg, Germany
[5]Deltares, Delft, Netherlands
[6]Department of Meteorology, University of Reading, Reading, United Kingdom
[7]National Centre for Atmospheric Science, University of Reading, Reading, United Kingdom
[8]Indian Institute for Tropical Meteorology, Pune, India

*Correspondence to*: Giorgia Di Capua (dicapua@pik-potsdam.de)

**Abstract.** Tropical convective activity represents a source of predictability for mid-latitude weather in the Northern Hemisphere. In winter, the El Niño–Southern Oscillation (ENSO) is the dominant source of predictability in the tropics and extra-tropics, but its role in summer is much less pronounced and the exact teleconnection pathways are not well understood. Here, we assess how tropical convection interacts with mid-latitude summer circulation at different intraseasonal timescales and how ENSO affects these interactions. First, we apply maximum covariance analysis (MCA) between tropical convective activity and mid-latitude geopotential height fields to identify the dominant modes of interaction. The first MCA mode connects the South Asian monsoon with the mid-latitude circumglobal teleconnection pattern. The second MCA mode connects the western North Pacific summer monsoon in the tropics with a wave-5 pattern centred over the North Pacific High in the mid-latitudes. We show that the MCA patterns are fairly insensitive to the selected intraseasonal timescale from weekly to 4-weekly data. To study the potential causal interdependencies between these modes and with other atmospheric fields, we apply the causal discovery method PCMCI at different timescales. PCMCI extends standard correlation analysis by removing the confounding effects of autocorrelation, indirect links and common drivers. In general, there is a two-way causal interaction between the tropics and mid-latitudes but the strength and sometimes sign of the causal link are timescale dependent. We introduce causal maps that show the regionally specific causal effect from each MCA mode. Those maps confirm the dominant patterns of interaction and in addition, highlight specific mid-latitude regions that are most strongly connected to tropical convection. In general, the identified causal teleconnection patterns are only mildly affected by ENSO and the tropical-mid-latitude linkages remain similar. Still, La Niña strengthens the South Asian monsoon generating a stronger response in the mid-latitudes, while during El Niño years, the Pacific pattern is reinforced. This study paves the way for process-based validation of boreal summer teleconnections in (sub-)seasonal forecast models and climate models and therefore works towards improved sub-seasonal predictions and climate projections.

## 1 Introduction

Tropical – mid-latitude teleconnections in boreal summer can have a great impact on surface weather conditions in the northern mid-latitudes (Ding and Wang, 2005; O'Reilly et al., 2018; Wang et al., 2001). Still, the  direct influence of the El Niño-Southern Oscillation (ENSO) on the mid-latitude circulation is weaker in summer than in winter (Branstator, 2002; Schubert et al., 2011; Thomson and Vallis, 2018). Instead, in summer, convective activity related to the Northern Hemisphere tropical monsoon systems can profoundly influence surface weather conditions in the mid-latitudes (Branstator, 2014; Ding and Wang, 2005; O'Reilly et al., 2018; Rodwell and Hoskins, 1996). Vice versa, mid-latitude wave trains and cyclonic activity at intraseasonal timescales can modulate the tropical monsoons, and have been linked to extreme rainfall events in the Indian region (Lau and Kim, 2011; Vellore et al., 2014, 2016). Therefore, tropical and mid-latitude regions are likely connected in

complex, two-way, teleconnection patterns operating at a range of sub-seasonal timescales (Di Capua et al., 2020; Ding and Wang, 2005, 2007).

During boreal summer, the South Asian monsoon (SAM), represents one of the most powerful features of the tropical/subtropical circulation. Characterized by heavy rainfall over central India and the Bay of Bengal, the SAM has strong intraseasonal variability associated with alternating active and break phases, linked to the boreal summer intraseasonal oscillation (BSISO, Choudhury and Krishnan, 2011; Gadgil and Joseph, 2003; Goswami et al., 1998; Krishnamurti and Surgi, 1987; Krishnan et al., 2000; Rao, 1976; Saha et al., 2012; Suhas et al., 2012). The western North Pacific summer monsoon (WNPSM) represents the Pacific counterpart to the SAM and is identified by strong rainfall over the sub-tropical western North Pacific (Li and Wang, 2005). Similar to the SAM, the WNPSM also exhibits strong intraseasonal oscillations (Wang and Xu, 1997).

Latent heat release due to strong monsoonal rainfall can influence subtropical and mid-latitude regions via Rossby wave teleconnections. The SAM has been connected to subtropical arid conditions in the North African region via the so-called monsoon – desert mechanism, creating reinforced descending motions over the Sahara during strong SAM phases (Rodwell and Hoskins, 1996; Stephan et al., 2019). This mechanism is fairly well captured by both  climate (Cherchi et al., 2014) and seasonal forecast models (Beverley et al., 2019).The SAM is also connected to mid-latitude circulation via its interaction with the circumglobal teleconnection pattern (CGT), a wave pattern with 5 centres of action encircling the northern mid-latitudes and affecting temperature and precipitation there (Ding and Wang, 2005; Kripalani et al., 1997). This wave-5 like CGT pattern can be identified through interannual to intraseasonal (weekly) timescales and it is likely connected with the SAM via two-way causal links (Di Capua et al., 2020). Seasonal forecast models are biased in their representation of the CGT, with typically a too weak CGT signal (Beverley et al., 2019). Therefore, seasonal forecasts miss an important source of predictability on intraseasonal timescales, primarily in summer (Weisheimer and Palmer, 2014).

The WNPSM has been shown to influence precipitation anomalies over North America via its relation to the Western Pacific – North America (WPNA) pattern (Wang et al., 2001). The WNPSM is shown to be related to surface pressure conditions over East Asia, with high pressure anomalies during years characterized by stronger WNPSM activity (Nitta, 1987). The WNPSM area also represents a genesis region for tropical cyclones in the North Pacific (Briegel and Frank, 1997). The WNPSM is weaker during the decaying phase of El Niño (Wang et al., 2001) and its related circulation anomalies provide a link from ENSO to the East Asian summer monsoon (Yim et al., 2008). Thus, in summary, the SAM appears particularly important for sub-seasonal variability over Eurasia, while the WNPSM is important for the Pacific-North American sector.

ENSO, operating at interannual timescales, might primarily influence the mid-latitude circulation via its effect on the SAM strength (Ding et al., 2011). During boreal summers preceding La Niña phases, a strengthening of the Walker circulation can enhance SAM rainfall, while El Niño phases often have an opposite effect (Joseph et al., 2011; Ju and Slingo, 1995; Terray et al., 2003; Wu et al., 2012). However, this relationship depends on the longitudinal position of the strongest El Niño related sea surface temperature (SST) anomalies (Krishna Kumar et al., 2006) and potentially has weakened over recent decades (Chakraborty and Krishnamurti, 2003; Krishna Kumar et al., 1999; Srivastava et al., 2017; Xavier et al., 2007). At interannual

timescales, anomalous tropical convection in the central-eastern Pacific related to ENSO has also been shown to affect mid-latitude circulation over the Euro-Atlantic sector as well as temperature and precipitation anomalies over Europe during boreal summer (O'Reilly et al., 2018). Trends in tropical SSTs play a crucial role in the interdecadal changes of this tropical-extratropical teleconnection (O'Reilly et al., 2019).

In general, a major challenge faced by teleconnection research is to understand the underlying physical processes and associated cause-effect relationships. Past observational studies have typically employed correlation analysis or linear regression techniques. Such analyses can however be dominated by spurious correlations and therefore can give only limited insight into cause-effect relationships. On the other hand, model-based studies can be affected by biases in the representation of circulation and precipitation characteristics (Beverley et al., 2019; Schubert et al., 2011; Weisheimer and Palmer, 2014),

which can feed back on each other. Also, although perturbation and sensitivity experiments can point towards potential causal relationships, they do not necessarily reveal the causal links between tropical and mid-latitudinal features, since the uncovered relationship may not be the dominant one.

In recent years, several approaches have been applied to identify causal relationships in climate and atmospheric sciences

(Runge et al., 2019b), ranging from Granger causality (McGraw and Barnes, 2018, 2020; Samarasinghe et al., 2019) to causal (Bayesian) graphical models (Ebert-Uphoff and Deng, 2012a, 2012b; Horenko et al., 2017; Pearl, 2000) and conditional independence-based network discovery methods for time series (Runge et al., 2019a). These studies have shown the ability of causal discovery tools to improve our understanding of several atmospheric circulation interactions such as Arctic – mid-latitudes connections (McGraw and Barnes, 2020; Samarasinghe et al., 2019), synoptic-scale disturbances between boreal

summer and boreal winter (Ebert-Uphoff and Deng, 2012a) and the relationship between ENSO and surface temperature in the American continent (McGraw and Barnes, 2018).

Here, we use a causal inference approach to study the relationships between the Northern Hemisphere mid-latitudes and the tropical belt during boreal summer at different intraseasonal timescales. We apply a causal discovery approach making use of

125 the so-called PCMCI (Peter & Clark algorithm combined with the Momentary Conditional Independence approach, see Section 2.3) method (Runge et al., 2019a) and then estimate physically interpretable causal links weights by (standardized) multivariate regression. The resulting weighted network representation of causal interdependencies is referred to as a Causal Effect Network (CEN) (Kretschmer et al., 2016). timescale Expanding our understanding of the corresponding physical mechanisms has the potential to improve seasonal and subseasonal forecasts in boreal summer. The main advantage of causal discovery tools is

130 that they can identify and remove spurious correlations (Runge et al., 2015b; Runge, 2018; Runge et al., 2019a) and thus provide insight into the potential causal relationships (McGraw and Barnes, 2018; Runge et al., 2014). Building upon this advanced methodology, we introduce a new concept called causal maps, visually highlighting causally related spatial structures. Finally, we assess the role of the ENSO background state on the identified causal relationships between the tropical belt and the mid-latitude circulation. The remainder of this paper is organized as follows: Section 2 presents the data and

methods used in this analysis. Section 3 describes the results obtained by applying CEN and causal maps to the identified research questions. Section 4 provides a discussion of the obtained results in the context of the existing literature and finally, Section 5 presents a short summary and conclusions.

## 2 Data and Methods

### 2.1 Data

In our analysis, we diagnose monsoon characteristics and Northern Hemisphere circulation features using outgoing longwave radiation (OLR) at the top of the atmosphere, geopotential height at 200 hPa (Z200) and 2m surface temperature (T2m) data from the ERA-Interim Reanalysis (Dee et al., 2011) for the period 1979-2018 (1.5°x1.5°). Strong tropical convection is characterized by high cloud tops and thus by low emission temperatures, which in turn correspond to low OLR values (Krishnan et al., 2000). In the tropical belt, OLR can be used as a proxy of convective activity, and therefore, rainfall. To select different ENSO phases, we use the monthly Niño3.4 index from NOAA (data are available at https://www.esrl.noaa.gov/psd/gcos_wgsp/Timeseries/Nino34/), representing the central Pacific SST anomalies. El Niño and La Niña events are discerned by periods of December-to-February Niño3.4 index values larger than 0.5°C or smaller than -0.5°C respectively. Then, we identify El Niño summers as those preceding the El Niño peak in winter and La Niña summers as those preceding the La Niña peak in winter. We use the Niño3.4 index since it has been shown to have a relatively strong connection to Indian monsoon rainfall (Krishna Kumar et al., 2006).

We also use the BSISO index as defined by Kikuchi et al. (2012) and Kikuchi and Wang (2010) (data are available at http://iprc.soest.hawaii.edu/users/kazuyosh/ISO_index/data/BSISO_25-90bpfil_pc.txt) in order to describe the phase and amplitude of the BSISO characterising the large-scale driver of active and break events over India. Causal discovery tool techniques require detrended anomalies centred at zero. Therefore, all data are linearly detrended and anomalies are calculated relative to an individual year's mean seasonal state by removing both the mean seasonal cycle and the year's mean seasonal state (i.e. the seasonal average from May to September, MJJAS) (Di Capua et al., 2020; Ding and Wang, 2007). Removing the year's mean seasonal state, and thus excluding the influence of interannual variations of the involved mechanisms, is essential to analyse intraseasonal variability of atmospheric components that present a strong interannual variably, such as the SAM.

### 2.2 Maximum covariance analysis

To extract the dominant co-variability patterns reflecting interactions between mid-latitude circulation in the Northern Hemisphere and tropical convection at intraseasonal timescales, we follow Ding et al. (2011) and apply maximum covariance analysis (MCA) to OLR fields (used as a proxy for convective activity) in the tropical belt (15°S-30°N, 0°-360°E) paired with Z200 fields in the northern mid-latitudes (25°N-75°N, 0°-360°E).

MCA identifies the patterns that explain the *greatest squared covariance between two different fields* (Ding et al., 2011; Wiedermann et al., 2017) and ranks them according to their explained squared covariance fraction (SCF) (Wilks, 2011). Among the available correlation based methods to highlight strong co-variability and reduce the dimensionality of a spatiotemporal dataset, MCA allows identification of patterns in pairs of variables that evolve simultaneously and may be causally related (via e.g. dynamical coupling between multiple climatological fields). MCA detects patterns that can explain

shared covariance, which cannot be achieved using other dimensionality reduction methods that consider individual variables separately, such as empirical orthogonal function (EOF) analysis. However, for providing a complete picture we will also discuss the corresponding EOF patterns and the fraction of variance explained for comparison with our MCA results.

Each MCA mode provides two coupled (2D) spatial patterns (one for tropical OLR and one for mid-latitude Z200) and two associated (1D) time series (the time-dependent MCA scores or pattern amplitudes for both fields), describing the magnitude

(prominence) and phase (sign) of those patterns for each time step. These (1D) time series are obtained by calculating the scalar product between each MCA spatial pattern (2D field) and the original spatial field of the associated variable at each time step as

$$A = \boldsymbol{u}^T \boldsymbol{X} \tag{1}$$

$$B = \boldsymbol{v}^T \boldsymbol{Y} \tag{2}$$

where $A$ and $B$ represent the two MCA scores for Z200 and OLR, $X$ and $Y$ are two matrices representing the Z200 and OLR fields, $\boldsymbol{u}$ and $\boldsymbol{v}$ are the coupled patterns that maximize their covariance $c$, defined as:

$$c = cov[A, B] = cov[\boldsymbol{u}^T \boldsymbol{X}, \boldsymbol{v}^T \boldsymbol{Y}] = \frac{1}{n-1}[\boldsymbol{u}^T \boldsymbol{X}(\boldsymbol{v}^T \boldsymbol{Y})^T] = \boldsymbol{u}^T \boldsymbol{C}_{xy} \boldsymbol{v} \tag{3}$$

and

$$\boldsymbol{C}_{xy} = \frac{1}{n-1} \boldsymbol{X} \boldsymbol{Y}^T \tag{4}$$

with $n$ denoting the number of observation times.

Here, we select the first two MCA modes that represent the dominant patterns of co-variability between tropical convection and mid-latitude circulation, and calculate time series for each MCA mode. These time series will be used as inputs for the causal discovery algorithm (see sections 2.3 and 2.4).

## 190 2.3 PCMCI and Causal Effect Networks

PCMCI is a causal discovery method based on the PC algorithm (named after its inventors Peter and Clark, see Spirtes et al., 2000) combined with the Momentary Conditional Independence approach (MCI, Runge et al., 2019). Given a set of univariate time series (called *actors*), PCMCI estimates their time series graph representing the conditional independencies among the time-lagged actors. In the context of the present work, actors are user-selected based on theoretical knowledge to represent

either a specific component of the atmospheric circulation or surface conditions estimated with MCA ($A$, $B$) or an individual grid point time series $C(lat,lon)$. Assuming linear dependencies, PCMCI uses partial correlations to iteratively test conditional

independencies and remove spurious links arising from autocorrelation effects, indirect links, or common drivers. For example, if an actor $Z$ drives $X$ at lag -1 and $Y$ at lag -2, then $X$ and $Y$ will be correlated, but the partial correlation $\rho(X_{\tau-1}, Y_t \mid Z_{\tau-2})$ will be zero. PCMCI efficiently conducts partial correlation tests to identify which links cannot be explained by other time-lagged actors. Compared to the standard PC algorithm, PCMCI better deals with autocorrelation and high-dimensional sets of actors (Runge et al., 2019a). The output of PCMCI is a $p$-value for each time-lagged causal link.

It is important to note that the term causal rests on specific assumptions (Runge, 2018; Spirtes et al., 2000), most importantly that it should be understood as "causal relative to the set of analysed actors". Therefore, adding (or removing) an actor can alter the result of PCMCI, highlighting the importance of having an expert-guided hypothesis underlying the choice of the selected set of actors. In addition, using partial correlation for a conditional independence test implies further assumptions such as the stationarity and linearity of the relationships. To control for multiple testing among the multiple grid locations in causal maps, we apply a false discovery rate (FDR) correction (Benjamini and Hochberg, 1995).

Based on the reconstructed network among the actor variables (at some significance level α), we determine the causal parents as the incoming links to each actor ($C(lat,lon)$, $A$, $B$), which can come from the pasts of $A$, $B$, or $C(lat,lon)$, i.e., { $A_{\tau=-1}$, $B_{\tau=-1}$, $C(lat,lon)_{\tau=-1}$, ... , $A_{\tau=-\tau max}$, $B_{\tau=-\tau max}$, $C(lat,lon)_{\tau=-\tau max}$}. In this analysis, $A$ and $B$ represent the two MCA scores obtained for a selected MCA mode, while $C(lat,lon)$ represents the grid point time series of a 2D field, e.g. T2m or Z200. In its first step, PCMCI iterates through partial correlations with increasing cardinality of conditions to remove the influence of common drivers and indirect links and estimate a preliminary set of parents. The first iteration of PC (cardinality 0) calculates the correlation between a selected time series, e.g. $A_{\tau=0}$, and the past of any other available time series, { $A_{\tau=-1}$, $B_{\tau=-1}$, $C(lat,lon)_{\tau=-1}$, ... , $A_{\tau=-\tau max}$, $B_{\tau=-\tau max}$, $C(lat,lon)_{\tau=-\tau max}$}, including its own past $A_{\tau=-1, ..., -\tau max}$. For illustration purposes, we here provide an example for $C(lat,lon)$, where $\rho$ denotes the correlation and $\tau$ is the lag that is being used in the network (in this example, $\tau_{max} = -2$):

$$\rho(C(lon,lat)_{\tau=o}, A_{\tau=-1}) = 0.32, p = 0.01 \tag{5}$$
$$\rho(C(lon,lat)_{\tau=o}, A_{\tau=-2}) = 0.13, p = 0.1$$
$$\rho(C(lon,lat)_{\tau=o}, B_{\tau=-1}) = 0.35, p = 0.005$$
$$\rho(C(lon,lat)_{\tau=o}, B_{\tau=-2}) = 0.23, p = 0.058$$
$$\rho(C(lon,lat)_{\tau=o}, C(lon,lat)_{\tau=-1}) = 0.41, p = 0.01$$
$$\rho(C(lon,lat)_{\tau=o}, C(lon,lat)_{\tau=-2}) = -0.16, p = 0.06$$

Applying a significance level α = 0.05, only three actors are significantly correlated with $C(lat,lon)$ at the chosen time lag. These form the initial preliminary set of parents for $C(lat,lon)$ and are ordered by the strength of their correlation:

$$P^0_{C(lon,lat)} = \{C(lat,lon)_{\tau=-1}, B_{\tau=-1}, A_{\tau=-1}\} \tag{6}$$

Next, partial correlations between $C(lat,lon)$ and each actor in $P^0_{C(lon,lat)}$ are calculated by conditioning on the strongest preliminary parent:

$$\rho(C(lat,lon)_{\tau=o}, C(lat,lon)_{\tau=-1} \mid B_{\tau=-1}) = 0.35, p = 0.02 \tag{7}$$

$$\rho(C(lat, lon)_{\tau=o}, B_{\tau=-1}|C(lat, lon)_{\tau=-1}) = 0.28, p = 0.03$$

$$\rho(C(lat, lon)_{\tau=o}, A_{\tau=-1}|C(lat, lon)_{\tau=-1}) = 0.25, p = 0.04$$

Parents with significant partial correlations will enter the second set of preliminary parents:

$$P^1_{C(lat,lon)} = \{C(lat, lon)_{\tau=-1}, B_{\tau=-1}, A_{\tau=-1}\} \tag{8}$$

Next, the partial correlation is calculated conditioning on the two strongest parents:

$$\rho(C(lat, lon)_{\tau=o}, C(lat, lon)_{\tau=-1}|B_{\tau=-1}, A_{\tau=-1}) = 0.31, p = 0.03 \tag{9}$$

$$\rho(C(lat, lon)_{\tau=o}, B_{\tau=-1}|C(lat, lon)_{\tau=-1}, A_{\tau=-1}) = 0.23, p = 0.04$$

$$\rho(C(lat, lon)_{\tau=o}, A_{\tau=-1}|C(lat, lon)_{\tau=-1}, B_{\tau=-1}) = 0.12, p = 0.08$$

Since it is not possible to further increase the dimension of the condition set, from the PC step, the preliminary parents converge to:

$$P^2_{C(lon,lat)} = \{C(lat, lon)_{\tau=-1}, B_{\tau=-1}\} \tag{10}$$

By repeating this step for each variable (and for each longitude and latitude position), preliminary sets of parents are estimated. Let's assume that in our example we also obtain:

$$P^3_A = \{ B_{\tau=-1}, A_{\tau=-2,}\} \tag{11}$$

$$P^2_B = \{ B_{\tau=-1}\}$$

In the MCI step, partial correlation is calculated again between each pair of actors (at different time lags) conditional on the above estimated sets of preliminary parents, whereby both sets of parents are conditioned upon. To give one example, this would lead to:

$$\rho\big(C(lat, lon)_{\tau=o}, A_{\tau=-1}\big|P^2_{C(lat,lon)}, P^3_A\big) =$$

$$= \rho(C(lat, lon)_{\tau=o}, A_{\tau=-1}|C(lat, lon)_{\tau=-1} \, B_{\tau=-1}, B_{\tau=-2}, A_{\tau=-3}) = 0.1, p = 0.3 \tag{12}$$

Note that the parents of $A_{\tau=-1}$ are shifted in time by $\tau = -1$. After repeating (12) for each pair of actors shown in (5) and for time lags from 0 to $\tau_{max}$, those parents that are significant in the MCI test will then form the final set of *causal parents* for each actor. We refer to Runge et al. (2019a) for a more detailed discussion and explanation of the algorithm design and extensive numerical experiments.

Finally, we estimate the Causal Effect Network (CEN) (Kretschmer et al., 2016; Runge et al., 2015a) among *A*, *B* and *C(lat,lon)* by applying standardized multiple regression of each actor onto its causal parents identified via PCMCI, i.e., for $Y \in$ in $A_t$, $B_t$, *C(lat,lon)*$_t$ and the parents *P*:

$$Y_t = \sum_i \beta_i X_i + \eta_Y \tag{13}$$

where $X_i \in P\{Y\}$, $i = 1, .., N$, i.e. the set of *N* parents of $Y_t$ and $\eta_Y$ is the residual of $Y_t$ (i.e. the difference between the observed value $Y_t$ and the value obtained by the linear regression on the causal parents $\sum_i \beta_i X_i$). Note that there can be different numbers *N* of parents for each actor.

Finally, the strength of a causal link $X_{t-\tau} \rightarrow Y_t$ is expressed in terms of the path coefficient $\beta$, which can be interpreted as the change in the expectation of $Y_t$ (in units of its standard deviation (s.d.)) induced by raising $X_{t-\tau}$ by 1 s.d., while keeping all other parents of $Y_t$ constant. Thus, for $\beta = 0.5$, a change in a causal parent of 1 s.d. at lag -1 corresponds to a change 0.5 s.d. in the analysed actor at lag 0 (Runge et al., 2015a). The influence of an actor on itself is referred to as the autocorrelation path coefficient, which must not be confused with the Pearson autocorrelation. A detailed description of the PCMCI algorithm is available in Runge et al. (2019), while recent applications can be found in Kretschmer et al. (2016, 2018) and (Di Capua et al., 2019, 2020).

## 2.4 Causal maps

To explore the causal effects that a specific actor has on a selected 3D (lat,lon, time) atmospheric field, we introduce the concept of *causal maps*. Conceptually, causal maps are similar to correlation maps, as they show the spatial pattern of the relationship between a 3D climate data set (covering two spatial dimensions plus time) and a 1D time series. However, instead of computing correlations between the time variations at each grid point and *one* additional time series, we apply here the PCMCI+CEN approach with actors consisting of the two MCA scores time series ($A$, $B$) and each individual grid point time series ($C(lat,lon)$). The causal map then shows the path coefficient $\beta$ from one of the MCA scores (as one actor) to this gridpoint, conditioned on all remaining actors. For a set of two actor timeseries ($A$ and $B$ in Fig. 1) and one time-varying atmospheric field $C$, we can thus derive two causal maps: one from $A$ to $C(lat,lon)$ conditioned on $B$ and on the autocorrelation in all actors, and one from $B$ to $C(lat,lon)$ conditioned on $A$ and on all autocorrelation effects. Figure 1 provides an illustrative example of this type of analysis. Both correlation maps (Fig. 1a) indicate a positive value for a specific geographical location highlighted with the black diamond. The CEN constructed for $A$, $B$ and $C(lat,lon)$ at this gridpoint is plotted in Fig. 1b and shows that only $B$ is causally connected to $C$. The correlation between $A$ and $C$ is thus due to an indirect link via $B$ (or to a common driver not included in the CEN). This is also seen in the causal maps showing the path coefficient $\beta$ which for the $B \rightarrow C(lat,lon)$ link is positive (right panel) but is non-significant for the $A \rightarrow C(lat,lon)$ link (left panel). In causal map visualization we can directly illustrate the effect of a specific actor on a global field (taking into account the influence of autocorrelation), indirect links and common driver effects due to other competing variables.

Here, we will derive causal maps using the time series obtained with MCA for modes 1 and 2 and Z200, OLR and T2m fields both for the entire time period (1979-2018) and for two subsets depicting different ENSO phases, to assess how the ENSO background state influences the causal relationships. El Niño (La Niña) summers are defined as summers preceding the El Niño (La Niña) peak in boreal winter. We thus obtain 14 La Niña years and 13 El Niño years (see Table 1 in the Supplementary material for a list of corresponding years and Fig. S1 for the associated SST anomaly composites). Although the strongest SST anomalies related to the ENSO phase are found in winter, warm (cold) SST patterns related to El Niño (La Niña) phases are already clearly developed during the preceding summers.

Finally, to test the robustness of our causal maps to the choice of time period, we calculate causal maps for a range of sub-periods. In 10 trials we removed 10% of the record (4 years). For ENSO-phase dependent causal maps, we have shorter time

series and we thus remove one year in each trial, leaving a set of 14 causal maps for La Niña events and 13 causal maps for El Niño events. As a result, we obtain an ensemble of causal maps and apply the false discovery rate correction to p-values of each single map. Then, both for the full period (1979-2018) and for El Niño and La Niña years separately, we masked out areas where less than 70% of the trials indicated a significant causal link, giving an indication of the robustness of our findings and at the same time suppressing noise.

A summary of the abbreviation and variable used in this analysis can be found in Table 1, while the parameters used for the PCMCI algorithm are reported in the Supplementary Material.

## 3 Results

### 3.1 Tropical – mid-latitude interactions: maximum covariance analysis

Figure 2a-d show the coupled patterns for the first two MCA modes between weekly tropical OLR and mid-latitude Z200. Figure 2e shows the associated time series of MCA scores for all four patterns (two for each MCA mode), obtained as explained in Section 2.2.

The first two MCA modes highlight the two key patterns of boreal summer monsoonal activity in the tropics along with the co-varying mid-latitude Z200 patterns. In both modes, the mid-latitudes are characterized by a zonally oriented circumglobal wave pattern with a wavenumber close to 5 (i.e. roughly 5 centres of action). However, the two wave patterns are phase shifted, aligned with the longitudinal position of the strongest monsoonal convection in the tropics.

The first MCA mode explains 18% of the squared covariance (squared covariance fraction, SCF) and shows a CGT-like wave-5 pattern in mid-latitude Z200. The Pearson correlation between the two time series of MCA scores for the first mode is r ~ 0.5. The spatial correlation with the weekly CGT pattern, as defined by Di Capua et al. 2020, is 0.52 (Fig. 2a). The CGT pattern also represents the second most important pattern in boreal summer mid-latitude circulation (Di Capua et al., 2020; Ding and Wang, 2005). This wave-5 pattern is linked to the South Asian monsoon (SAM) activity via its positive centre of action east of the Caspian Sea (see Fig. 2a). Applying MCA, we find that the CGT pattern co-varies with a band of enhanced tropical convective activity that extends from the Arabian Sea towards Southeast Asia, with a peak of convective activity over the Bay of Bengal (Fig. 2b) (Kang et al., 1999). Using OLR composites and the Kikuchi Boreal Summer Intraseasonal Oscillation (BSISO) index, we explicitly show that the temporal evolution of SAM convective activity as defined in Fig. 2b at weekly timescales closely resembles the evolution of the BSISO (Goswami and Ajaya Mohan, 2001; Saha et al., 2012) (see Figs. S2-S3 and further discussion in the Supplementary Material). Therefore, we explicitly link the region of low OLR identified in Fig. 2b over the northern Indian Ocean and the Indian subcontinent to SAM activity as described in the literature. Note that we name each MCA pattern after a characteristic regional feature, but the analysis is applied to the larger geographical domains as shown in Figure 2.

The second mode of co-variability explains a SCF of 14% and is characterized by a region of strong positive Z200 anomalies located at ~ 45° N, over the western North Pacific, directly to the west of the dateline (i.e. the most prominent centre of action

of the mid-latitude wave). The Pearson correlation between the two time series of MCA scores for the first mode is r ~ 0.6. We will refer to this pattern as the North Pacific High (NPH) (Fig. 2c). The NPH is the summer counterpart of the North Pacific subtropical high, which characterizes boreal winter. During summer, this high pressure region is displaced northward by the start of the monsoon season over the western Pacific Ocean and replaces the Aleutian Low (Lu, 2001; Riyu, 2002). The NPH is associated with a region of enhanced convection over the sub-tropical western North Pacific, related to the western North Pacific summer monsoon (WNPSM) convective activity (Fig. 2d) (Li and Wang, 2005; Nitta, 1987; Wang et al., 2001). The WNPSM core domain extends from 110°-160°E and 10°-20°N, while the boundary with the SAM is located over the South China Sea (Murakami and Matsumoto, 1994). The WNPSM is characterized by a late sudden onset (end of July) and a peak in rainfall activity during August and September, which is different from the SAM that features an earlier onset (in June) and peak rainfall activity during July-August.

We compare the patterns obtained with MCA with those obtained with EOF analysis of Z200 and OLR fields (see Fig. S4 in the Supplementary Material). We find that the closest match of the Z200 MCA mode 1 pattern is with Z200 EOF 2 (spatial correlation ~ 0.8), while the closest match of Z200 MCA mode 2 is with EOF 1 (spatial correlation ~ 0.6). OLR MCA mode 1 has the closest match with EOF 2 (spatial correlation ~ 0.5), while OLR MCA mode 2 has the closest match with EOF 5 (spatial correlation ~ 0.4). Thus, in general our MCA patterns also reflect the first two EOFs of Z200 and OLR indicating that they explain an important fraction of the regional variability. Nevertheless, here we are interested in those patterns that can explain shared covariance, which cannot be achieved by using EOF analysis alone. Therefore, we use the MCA-defined patterns for the following part of the analysis.

We also investigate whether the obtained MCA patterns are sensitive to the choice of OLR in representing tropical convective activity. Using vertical velocity, another proxy for tropical convection where strong convective activity is represented by enhanced upward motion, shows qualitatively the same patterns as those in Fig. 2a-d (see Fig. S5 in the Supplementary Material). When velocity potential is used instead of OLR, the first MCA mode still closely resembles the OLR/Z200 MCA mode 1, while the second MCA mode only partly captures features in the western Indian Ocean (see Fig. S6 in the Supplementary Material).

Application of MCA to 4-weekly data gives nearly identical MCA patterns but with somewhat lower magnitude of the Z200 and OLR anomalies (see Fig. S7 in the Supplementary Material). In this case, we define both 4-weekly and weekly MCA scores by projecting 4-weekly MCA patterns onto 4-weekly and weekly data respectively (see Fig. S7e-f in the Supplementary Material). In this way, we check whether the analysis is robust given different definitions of the MCA patterns.

### 3.2 Influence of tropical – mid-latitude MCA modes on Northern Hemisphere circulation

To show how each MCA mode affects the circulation and surface conditions in the Northern Hemisphere, we calculate causal maps for the influence of SAM, CGT, WNPSM and NPH time series (as defined in Fig. 2e) on selected atmospheric fields in the Northern Hemisphere (15°S-75°N, 0°-360°E). Although we use $\tau_{max}$ = -2 and $\tau_{min}$ = 0, we plot only $\beta$ values for lag -1 (week), as $\beta$ values for longer time lags are mostly nonsignificant. This way the past behaviour of each actor, with potential

confounding effects, is also accounted for in the corresponding grid-point CEN. Note that we only show robust links as defined in Section 2.4 and the masks used to plot the results are shown in Figs. S8-S9 in the Supplementary Material.

Figure 3 shows the causal maps for weekly Z200, OLR and T2m fields with SAM and CGT time series, including correlation maps for weekly Z200 fields with SAM and CGT time series. Referring to the schematic illustrated in Fig. 1 and following
the PCMCI algorithm explanation (section 2.3), here the *A* and *B* time series are represented by the SAM and CGT time series respectively, while *C(lat,lon)* is represented by Z200, OLR and T2m fields. In the mid-latitudes, the correlation map between Z200 and SAM (Fig. 3a) shows some similarities in the correlation map between Z200 and CGT (Fig. 3b), with negative correlation regions over central Europe and Scandinavia (Region 2 and Region 4) and over the eastern North Pacific and eastern Canada visible in both plots (regions 3 and 6). Both correlation maps also display a positive correlation over northern
Africa (Region 1), though with smaller values in the CGT plot. The causal map for the link SAM $_{\tau = -1}$ → Z200 $_{\tau = 0}$ (after removing the effects of the CGT and of the past of both SAM and Z200) shows that the path coefficient $\beta$ remains pronounced over northern India and northern Africa (Region 1 in Fig. 3c), with values $\beta \sim 0.3\text{-}0.4$. Interestingly, those regions disappear completely in the causal map for the link CGT $_{\tau = -1}$ → Z200 $_{\tau = 0}$ (after removing the effects from SAM and of the past of both CGT and Z200). Thus, in this way, we can separate the signal coming from SAM convective activity from signals originating
from the CGT pattern. Also, the causal maps in Figs. 3c and 3d indicate that the influence of SAM on Europe (negative path coefficients shown by Region 2 in Fig. 3c) and the North Pacific (seesaw of positive and negative path coefficients shown by Region 3 in Fig. 3c) is *not* mediated via the CGT. This influence is weaker than that over the North Africa, with values of $\beta \sim$ 0.1-0.3. In turn, the influence of SAM on other mid-latitude regions (over the North Atlantic, Region 4, over some parts of East Asia, Region 5 and Canada/USA, Region 6 in Fig. 3d) is mediated via the CGT, with values of $\beta \sim 0.1\text{-}0.3$.

In the mid-latitudes, the causal maps for OLR and T2m (Figs. 3e-h) are largely consistent with those obtained for Z200, with negative $\beta$ values for OLR representing wet anomalies overlapping with negative $\beta$ values for T2m representing colder anomalies and dry anomalies (positive $\beta$ values for OLR) overlapping with warm T2m (positive $\beta$ values for T2m). The CGT influence is mostly concentrated in the mid-latitude regions, where the same Regions 4 to 6 identified in Fig. 3d, can also be found in Figs. 3f and 3h. A significant and consistent negative causal effect of the CGT pattern on OLR values in the tropical
regions can only be seen in a small area in the western Indian Ocean (Region 1 in Fig. 3f). Again, the OLR and T2m causal maps indicate that the SAM has a direct influence on northern Africa and Europe as well as tropical Africa (Region 1 in Figs. 3e and 3g). Over the Indian peninsula and Indochina, strong convective motions (negative $\beta$ values for OLR in Fig. 3e) are accompanied by colder temperature (negative $\beta$ values for T2m in Fig. 3g), related to increased precipitation and consequently, decreased surface temperatures during active SAM activity. The influence of SAM on the western North Pacific identified by
Region 3 in Fig. 3c is also detected in OLR (Region 3 in Fig. 3e). Negative $\beta$ values found over Region 2 in Fig. 3c are only slightly visible in OLR and T2m. However, we should also remind the reader that our causal maps show only the most robust links                              (see                              Section                              2.4).

Figure 4 shows the same set of results but now for the second MCA mode consisting of WNPSM and NPH pattern related time series. Here, our *A* and *B* time series are represented by the WNPSM and NPH time series while *C(lat,lon)* is again represented by either Z200, OLR or T2m fields. As expected, both correlation maps resemble the Z200 field of MCA 2 (Fig. 2c,d) with two characteristic features: an arch-shaped wave pattern in the North Pacific (Regions 7,8 and 9 in Figs. 4a and 4b) with a prominent positive correlation over the NPH region and over western North America and two weaker centres of action over the Eurasian continent (Region 10 and 11 in Fig. 4b and Region 11 in Fig. 4a). The corresponding causal maps based on CENs are given in Figs. 4c (path coefficient $\beta$ for the link WNPSM $_{\tau = -1} \to$ Z200 $_{\tau = 0}$) and 4d (for the link NPH $_{\tau = -1} \to$ Z200 $_{\tau = 0}$). If we compare the correlation maps (Fig. 4a,b) with the causal maps (Fig. 4c,d), we find great similarity in the spatial structures of the Z200 patterns over the North Pacific in both figures (Region 7 in Fig. 4c,d) with $\beta \sim 0.1$-$0.3$ for the influence of NPH on Z200 and $\beta \sim 0.1$-$0.2$ for the influence of WNPSM on Z200. These causal maps show that the influence of the WNPSM on Z200 (after removing the effect of the NPH) is confined to the North Pacific alone (Regions 7 and 8 in Fig. 4c). The causal effect of the NPH pattern on Z200 (after removing effects of WNPSM) shows two most prominent regions displaying a significant positive path coefficient $\beta \sim 0.2$-$0.4$ over the NPH region (Region 7 in Fig. 4d) and over the US west coast (Region 9 in Fig. 4d). Region 7 in the Pacific sector coincides with that found for the WNPSM causal map (Fig. 4c). This suggests that the NPH is reinforced both by convective activity of the WNPSM and by the mid-latitude wave pattern localized in the North Pacific. Regions 10 and 11 found in Figs. 4a and 4b disappear in both Figs. 4c and 4d, showing that the correlation in these regions is mostly explained by Z200 activity in the mid-latitude itself (note that in the CEN we also condition on the past of Z200) or by other factors not considered in this analysis.

Next, we compute the causal maps for the influence of WNPSM and NPH on weekly OLR and T2m fields (Figs. 4e,f and 4g,h respectively). The impact of the WNPSM on OLR and T2m fields is very weak, though it is possible to recognise some negative $\beta \sim 0.1$-$0.2$ over the Arabian Sea and over the WNPSM area (Region 8 in Fig. 4e). The impact of NPH on OLR and T2m causal maps shows some similarities with the correlation map shown in Fig. 4b. T2m and OLR show the strongest effect over North America (Region 9 in Figs. 4f and 4h) with $\beta \sim 0.2$-$0.4$, and this result is largely consistent with that obtained for Z200, with positive Z200 anomalies hinting at warm and dry weather in the mid-latitudes related to an active WNPSM. Over Eurasia, it is possible to recognize Regions 10 and 11 in both ORL and T2m causal maps but with smaller regions and lower $\beta \sim 0.1$-$0.2$ (Figs. 4f and 4h). Thus, these results highlight the importance of the NPH pattern in shaping surface temperatures across the northern mid-latitudes.

Using weekly MCA scores obtained from 4-weekly MCA patterns (Fig. S7) gives consistent results, showing that the analysis is robust when a different definition of the MCA pattern is chosen (see Figs. S10-S11 in the Supplementary Material). Causal maps calculated for 4-weekly Z200 for both MCA 1 and MCA 2 show less significant results, likely due to the limited time series length (not shown).

### 3.3 The influence of ENSO on tropical – mid-latitude causal interactions

Next, we assess how the ENSO background state influences the causal relationships between mid-latitude and tropical patterns in boreal summer. We recalculate the causal maps for both MCA scores and Z200 fields, where $A$ and $B$ time series are represented by the SAM and CGT time series for the first MCA mode and by the WNPSM and NPH time series for the second MCA mode, while $C(lat,lon)$ is represented by Z200. As for Fig. 3 and 4, the robustness mask used to plot the results shown in Fig. 5 is shown in Fig. S12 in the Supplementary Material.

In general, the strength and the sign of the patterns seen in the causal maps (Fig. 3 and 4) are not markedly affected by ENSO, though we can see higher absolute values of $\beta \sim 0.1$ in Fig. 5 with respect to Figs. 3 and 4. During La Niña years, the effect of SAM on the Sahara Desert intensifies and also its effects on the Tibetan Plateau and in the mid-latitudes are more pronounced (Region 1, 2 and 6 in Fig. 5c). This is likely related to stronger SAM convective activity during La Niña summers. The region of negative causal effect of SAM on central Europe, is present only during La Niña summers (with a $\beta \sim 0.2$-0.3 and larger in area when compared to the 1979-2018 causal map in Fig. 3c), and it disappears during El Niño summers. This signal is possibly linked to the strong positive causal effect over the Sahara Desert (Region 1). The region of positive causal effect over the Tibetan Plateau shows the same intensity as for the 1979-2018 period ($\beta \sim 0.2$-0.3) though it remains more confined over the Indian subcontinent and the Tibetan Plateau with respect to the 1979-2018 period (Region 1 in Fig. 5c). During La Niña, we also see an area of positive β values over North America (Region 6 in Fig. 5c) that is not present during El Niño.

During El Niño summers, the influence of the SAM is almost completely absent in the mid-latitude regions, with only one region of low $\beta$ values over the eastern North Pacific still being present (Region 3 in Fig. 5a). The positive $\beta$ values over the tropical Pacific found in Fig. 3c disappear during La Niña and only some residues of this region are seen during El Niño years (Region 8 in Fig. 5a).

In the western North Pacific, the most notable feature is the presence of both the WNPSM and NPH on the North Pacific only during El Niño summers (Figs. 5e,f). During those summers, the positive causal effect of the WNPSM over the western North Pacific (Region 7 and 8 in Fig. 5e) intensifies in magnitude (absolute $\beta \sim 0.3$-0.4) with respect to the 1979-2018 mean pattern (Fig. 4c), although the geographical extent of Region 7 shrinks. Over the western tropical Pacific, in correspondence with the La Niña warm pool, a region of positive causal effect is shown (Region 8 in Fig. 5e). These features disappear during La Niña summers. Thus, the second MCA mode (the WNPSM-NPH pair) has its strongest effect during El Niño summers, whereas the first MCA mode (SAM-CGT pair) is important during both La Niña and El Niño summers but with different spatial characteristics.

Calculating MCA pattern during different ENSO phases does not change the results in a qualitative way, although the order of the patterns is reversed in La Niña summers (see Fig. S13, in the Supplementary Material). Moreover, we have checked whether the distribution of the spatial correlation between each MCA mode and the respective Z200/OLR fields changes during different ENSO phases and found that ENSO does not affect the frequency of each pattern in a significant way (see Fig. S14, in the Supplementary Material).

As for the 1979-2018 causal maps, when weekly MCA scores obtained from 4-weekly MCA patterns (Fig. S7) are used to provide ENSO-dependent causal maps, consistent results are obtained (see Figs. S15 in the Supplementary Material). Further analysis of possible physical explanations is provided in Section 4.

### 3.4 MCA causal interactions

Finally, we study the role of timescales on the causal interaction patterns presented above. We create CENs between the two time series of scores for each MCA mode, as identified in Fig. 2 and Fig. S7, and do so for weekly and 4-weekly data for the 1979-2018 period. Figure 6 plots the path coefficient $\beta$ for two separate sets of CENs built for MCA mode 1 (SAM with CGT, Fig. 6a) and MCA mode 2 (WNPSM with NPH, Fig. 6b) for both 4-weekly and weekly timescales. As for the causal maps, we use $\tau_{max} = -2$ for weekly data and $\tau_{max} = -1$ for 4-weekly data. In both cases, $\tau_{min} = 0$.

At the 4-weekly timescale, the WNPSM-NPH pair does not show significant causal links (Figs. 6b). The SAM-CGT pair shows two fairly strong causal links with absolute values $\beta \sim 0.3\text{-}0.4$, though with different signs (Figs. 6a). The northward link, i.e. $SAM_{\tau=-1} \rightarrow CGT_{\tau=0}$, shows a positive $\beta \sim 0.4$: a 1 s.d. shift in the SAM leads to a $\sim 0.4$ s.d. positive shift in the CGT 4 weeks later (Fig. 6a). The southward link, i.e. $CGT_{\tau=-1} \rightarrow SAM_{\tau=0}$, shows $\beta \sim -0.3$, meaning that at this timescale a more intense CGT pattern leads to a weakening of the SAM pattern 4 weeks later (Fig. 6a).

At the weekly timescale, both the $WNPSM_{\tau=-1} \rightarrow NPH_{\tau=0}$ and the $NPH_{\tau=-1} \rightarrow WNPSM_{\tau=0}$ links show a $\beta \sim 0.1\text{-}0.2$, indicating that the two-way link has a similar magnitude in both southward and northward directions (Fig. 6b). At this timescale, the path coefficient $\beta$ for the $SAM_{\tau=-1} \rightarrow CGT_{\tau=0}$ link is about a factor 4 smaller than that for the 4-weekly timescale (Fig. 6a). The southward link, $CGT_{\tau=-1} \rightarrow SAM_{\tau=0}$, shows a positive $\beta \sim 0.2$ that is about twice as strong as the northward link (Fig. 6a). Thus, the influence of the SAM on the CGT pattern is weak (but present) at shorter (weekly) timescales, but much stronger at longer (4-weekly) timescales.

Finally, we tested how the CENs change when the 4-weekly signal is removed from the weekly time series: Each 4-weekly mean is removed from the four values of the corresponding weekly data (Fig. S16). This way, we attempt to isolate the dominant timescales of physical processes behind the different MCA patterns. This is similar in rationale to removing the effects of interannual variability before quantifying intraseasonal variability. Results for the first MCA (SAM and CGT) shows that the path coefficient $\beta$ for $CGT_{\tau=-1} \rightarrow SAM_{\tau=0}$ link remains almost unaffected (see Fig. S16a, in the Supplementary Material). This suggests that this southward link typically operates at weekly timescales (rather than 4-weekly), which is rather intuitive since mid-latitude variability dominates at synoptic timescales. In contrast, the path coefficient $\beta$ for the northward link ($SAM_{\tau=-1} \rightarrow CGT_{\tau=0}$) becomes insignificant when the 4-weekly signal is removed from the weekly time series, suggesting that the influence from the tropics via the SAM pattern operates at longer, 4-weekly timescales. Removing the 4-weekly signal from the weekly time series for the second MCA (WNPSM and NPH) roughly halves the path coefficient $\beta$ for both the northward and southward link (see Fig. S16b in the Supplementary Material).

## 4 Discussion

In our analysis, we have found that the dominant patterns of interaction between the tropics and mid-latitudes remain qualitatively similar across different sub-seasonal timescales (weekly and 4-weekly averages) (Fig. 2 and Fig. S7 in the Supplementary Material). Two pairs of co-varying patterns are identified: a) convective activity of the South Asian monsoon (SAM) paired with a mid-latitude wavenumber-5 wave train resembling the circumglobal teleconnection (CGT) pattern and b) convective activity of the western North Pacific summer monsoon (WNPSM) paired with a second wave-5 circumglobal wave pattern with its strongest action centre represented by the North Pacific High (NPH). This second mid-latitude wave pattern is phase shifted with respect to the CGT pattern, to the longitudinal position of WNPSM monsoonal convection in the tropics. These patterns of sub-seasonal co-variability between the mid-latitudes and tropics during boreal summer are remarkably similar to those identified by Ding et al. (2011) for interannual timescales. This consistency across timescales (from weekly over monthly to interannual) suggests that the interannual patterns originate from a summation of the same patterns at sub-seasonal timescales. Still, the strength and sign of the causal interactions are timescale dependent. At longer timescales (from monthly to seasonal) slowly varying components such as tropical SST and associated regions of convective activity dominate tropical – mid-latitude interactions. Therefore, on these timescales the causal links from the tropics to mid-latitudes tend to be stronger. At shorter (weekly) timescales, in general a two-way positive feedback between the tropics and mid-latitudes is found, although strong variability in the mid-latitudes dominates over the tropical convection and thus the reversed southward pathways become stronger (Fig. 6). Moreover, we have introduced a novel visualization approach – termed causal maps – that can provide regionally specific information on the causal influence of a specific teleconnection source, and how this signal gets mediated. In this way, we identify mid-latitude regions and surface weather conditions that are influenced by tropical drivers by taking into account the linear influence of both MCA patterns together (for each MCA mode). The strongest causal effect of SAM convection is found over the Saharan region, and depicts the monsoon-desert mechanism (Rodwell and Hoskins, 1996). Also important is the effect of SAM on the central Asian CGT centre of action (see Di Capua et al., 2020). The SAM also appears to directly influence geopotential heights in the North Pacific and central European surface temperatures one week ahead (Fig. 3c,e). The influence of the CGT pattern is stronger over the mid-latitude regions, nevertheless some influence on the Indian Ocean is detected (Fig. 3d,f), further supporting the results shown in Fig. 6. In the North Pacific there is a clear positive influence from the WNPSM towards the NPH patterns reflecting a Hadley cell-like circulation (Fig. 4d). The direct causal effect from the NPH on surface weather conditions is particularly strong in central North America while its direct effect back on the tropics is weak (Fig. 4d,f,h). Thus, for MCA mode 2, the signal from the WNPSM towards the NPH is consistent both in simple CEN (Fig. 6) and causal maps (Fig. 4), while the direct influence of the NPH on the tropical belt is present but weaker and less robust (see Fig S9 in the Supplementary Material).

In the tropical belt, the processes behind the identified MCA patterns are linked to strong convection related to monsoon activity. Though tropical convection is characterized by heavy precipitation with a typical duration of less than a day, the

latent heat release can act as a Rossby wave source for up to two weeks after the initial forcing is removed (Branstator, 2014).

Moreover, while individual convective events are short-lived, the regions of dominant convective activity in the tropics change on much longer timescales, such as in response to the BSISO (30-60 days). Thus, this finding could serve as a possible explanation for why the patterns identified at weekly and 4-weekly timescales show great similarity (see also the discussion surrounding Figs. S2 and S3 in the Supplementary Material). It appears reasonable to assume that the tropics operate at longer timescales providing potential sources of predictability at seasonal-to-subseasonal (S2S) timescales. In contrast, mid-latitude

circulation in summer is weaker than in winter and is characterized by circumglobal wave trains with typical timescales of about one or two weeks (Di Capua et al., 2020; Ding and Wang, 2007; Kornhuber et al., 2016).

On the western North Pacific side, our findings linking the WNPSM convective activity to the NPH, and in turn to a wave-5 circumglobal wave train that affects surface weather condition in the mid-latitudes, further support results from previous studies suggesting that convective activity related to this oceanic monsoon system can enhance the high pressure found in the

535 North Pacific mid-latitudes and that this affects weather conditions in western North America (Chou et al., 2003; Wang et al., 2001). Four centres of action over northern and eastern Europe, central Asia, central North America and the central North Pacific are identified in the T2m causal map (Fig. 4h). Eastern Europe and central North America were also identified to be teleconnected regions associated with a boreal summer wave-5 by Kornhuber et al. (2020), who highlighted the risk for potential bread-basket failures. Recent evidence indicate that land-atmosphere interactions and increased aridity in mid-latitude

regions such as North America and Europe may constitute an enhancing mechanism for the amplification of circumglobal quasi-stationary Rossby wave events during boreal summer (Teng and Branstator, 2019). Therefore, our results support the importance of the role of Pacific forcing for this wave-5 circumglobal wave pattern. Although other mechanisms could also be relevant in exciting and maintaining this pattern, the link to the WNPSM convection may hold the potential to affect seasonal forecasts and climate risks, such as heat waves and simultaneous crop failures.

We have applied our causal map analyses to specific ENSO phases to assess the role of El Niño and La Niña in modulating the interactions between mid-latitude circulation and tropical convection in boreal summer. These analyses suggest that in general the ENSO phase does not change the qualitative nature of the causal relationships between different MCA patterns: the signs and strengths of the causal links are largely unaffected (see Fig. 5). Moreover, the same MCA patterns occur both in

La Niña and El Niño summers, and their frequency is hardly affected (Figs. S13-S14). Nevertheless, during La Niña summers, the effect of the SAM-CGT mode is reinforced over Europe, North Africa and the Indian subcontinent and reaches northward towards Canada while during El Niño summers the effect of the SAM is mainly confined to the tropical belt. For the WNPSM-NPH pattern, a clear asymmetry between El Niño and La Niña summers is shown, with a stronger signal during El Niño years (Fig. 5e,f) that is absent during La Niña years. Although this effect is not very large, it is still significant. During La Niña

summers, SAM exerts a stronger causal effect on the Tibetan High than over the entire 1979-2018 period, along with a reinforced monsoon-desert mechanism and a stronger effect on European circulation. This could be because under La Niña conditions, the SAM circulation is supported by a favourable Walker circulation (Ju and Slingo, 1995; Terray et al., 2003).

The same applies to the southward link: although ENSO does not alter the standardized causal effect from the CGT to SAM, a stronger CGT pattern in the mid-latitudes would have a stronger absolute effect on SAM activity at the weekly timescale. At
interannual timescales, Ding et al. (2011) show that the SAM-CGT pair is strongest during the developing phase of ENSO. Therefore, our results further support the hypothesis that ENSO acts on the CGT pattern via its influence on SAM activity, in agreement with Ding et al. (2011). This finding is also in agreement with previous work showing that, at decadal timescales, the CGT and Silk Road patterns intensify under PDO negative (i.e. La Niña-like) forcing (Stephan et al., 2019). During El Niño summers, the SAM shows a more prominent effect on the tropical Pacific. Nevertheless, since we condition on the effect
of the CGT, we cannot exclude that this strong signal over the Niño-3.4 region may be due to an element outside our CEN. In the North Pacific, causal maps for different ENSO phases show stronger activity of both WNPSM and NPH links during El Niño summers, consistent with previous literature (Chou et al., 2003; Liu et al., 2016). During El Niño events, tropical convection shifts together with SST anomalies towards the central-eastern Pacific, which may favour WNPSM convective activity. In contrast, during La Niña summers convection is shifted towards the Maritime Continent and the western tropical
Indian Ocean, reducing convective activity over the central Pacific and WNPSM region. A weaker WNPSM system may in turn be more prone to the influence of mid-latitude variability on the NPH.

Future projections describe an increase in monsoon precipitation associated with increasing global mean temperature and thermodynamic arguments (Menon et al., 2013; Turner and Annamalai, 2012). Quantifying teleconnections between the tropics
and mid-latitudes is important in order to better understand and constrain future changes in boreal summer circulation, as uncertainty may arise due to changing connections to remote regions. While simulations show great uncertainty in the ENSO response to global warming (Cai et al., 2015; Chen et al., 2017a, 2015, 2017b), observations show a La Niña-like warming trend in central-western Pacific SST (Kohyama et al., 2017; Mujumdar et al., 2012). A better understating of these teleconnections in observations can help to improve S2S forecasts. Verifying the existence and strength of causal
teleconnections in forecast models could help diagnose the origin of model biases. E.g. one could disentangle whether lower forecast skill (such as in the mid-latitude regions in summer) is related to local processes or to a misrepresentation of remote drivers. Beverley et al. (2019) showed that the CGT representation in seasonal forecasts is too weak. The CGT is important for predictability of summer extremes and its relationship with the SAM may provide some information to improve predictability. Therefore, these methods could help answering the question "where do model biases come from?" and help
developing a physics-based bias correction. At the same time, CEN provide an encoded statistical predictive model, which can be used for actual forecasting (Di Capua et al., 2019; Kretschmer et al., 2017; Lehmann et al., 2020). Our analysis shows that at 4-weekly timescale, the effect of SAM on the CGT pattern has a path coefficient $\beta \sim 0.4$, thus indicating some potential for predictability. Future studies should analyse how the causal links between these teleconnection patterns are reproduced in corresponding state-of-the-art S2S forecast and climate models, respectively.

Finally, it should not be forgotten that in the context of the present work, *causal interpretation* rests upon several assumptions, such as the causal Markov condition, faithfulness, causal sufficiency, stationarity of the causal links and assumptions about the dependence-type (Runge, 2018). These assumptions can be violated in a real system and it is important to be aware of the associated typical challenges for causal discovery in Earth system sciences (Runge et al., 2019a). Causal sufficiency requires that all relevant actors in a specific system are accounted for. Here, given the limited set of actors analysed, we cannot rule out that other excluded actors may act as important (common) drivers. Therefore, the obtained links can be considered *causal* only with respect to the specific set of actors used here. However, the *absence* of a link can still be interpreted as a likely indication that no direct physical connection among the respective variables exists (Runge, 2018). Moreover, we assume linear dependencies and stationarity for the detection of causal links. While linearity has been shown to be a useful assumption in previous work (Di Capua et al., 2020), monsoon dynamics behaves partly nonlinearly and therefore, our causal networks only capture some part of the underlying mechanisms by construction. Also, the SAM teleconnections might well behave in an nonstationary manner on decadal timescales (Di Capua et al., 2019; Robock et al., 2003). We therefore cannot exclude the possibility that (multi-)decadal oscillations such as the Pacific Decadal Oscillation may influence our results. However, the amount of reliable data is limited and this prohibits the application of nonlinear measures or the study of effects due to nonstationarity.

## 5 Conclusions

We have analysed the interdependencies and spatial effects of the two main MCA modes of co-variability between tropical convection and Northern Hemisphere mid-latitude circulation in boreal summer. The first MCA pair connects the circumglobal teleconnection (CGT) pattern in the mid-latitudes with the South Asian monsoon (SAM) convection, while the second MCA pair connects the western North Pacific summer monsoon (WNPSM) convection with a second circumglobal pattern related to the North Pacific High (NPH). These patterns appear qualitatively independent of the analysed timescales and emerge in weekly, 4-weekly and interannual analyses. The strength of the causal links *is* timescale dependent. In particular, the influence of SAM on CGT is strongest at the 4-weekly timescale, while the reversed link is stronger at weekly timescale. The patterns and sign of the standardized causal effect links are also not strongly affected by ENSO. During La Niña years the effect of the SAM on the mid-latitudes intensifies, while we find statistically significant links for the WNPSM effect on the mid-latitudes only for El Niño years. Moreover, the boreal summer intraseasonal oscillation exerts strong control on the SAM convection at various lags.

Furthermore, we have introduced causal maps, a new application of the concept of causal effect networks and have highlighted how this method can overcome limitations of correlation maps by removing spurious links. These causal maps further confirm our findings by showing a general positive two-way causal relationship between the dominant modes. Moreover, they highlight

specific regions in the mid-latitudes that are particularly affected by the tropical modes (e.g. Eurasia, North America). These findings provide an improved understanding of the interactions between tropical convective activity and circumglobal wave

trains that characterize mid-latitude circulation in boreal summer. This may help improving sub-seasonal forecasts as well as constraining future projections of boreal summer circulation. Further work shall assess whether these causal relationships are captured by general circulation models and whether this knowledge can be used to improve seasonal forecasts over the mid-latitudes.

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

| ABBREVIATION | FULL NAME | DIMENSIONS |
|---|---|---|
| **BSISO** | Boreal summer intraseasonal oscillation | 1D time series |

| | | |
|---|---|---|
| **SAM** | South Asian monsoon - MCA mode 1 OLR | 2D spatial pattern + 1D time series |
| **CGT** | Circumglobal teleconnection pattern – MCA mode 1 Z200 | 2D spatial pattern + 1D time series |
| **WNPSM** | Western North Pacific summer monsoon – MCA mode 2 OLR | 2D spatial pattern + 1D time series |
| **NPH** | North Pacific High – MCA mode 2 Z200 | 2D spatial pattern + 1D time series |
| **Z200** | Geopotential height at 200 hPa | 2D field + time |
| **OLR** | Outgoing longwave radiation | 2D field + time |
| **T2M** | 2m temperature | 2D field + time |

**Table 1. Abbreviations.**

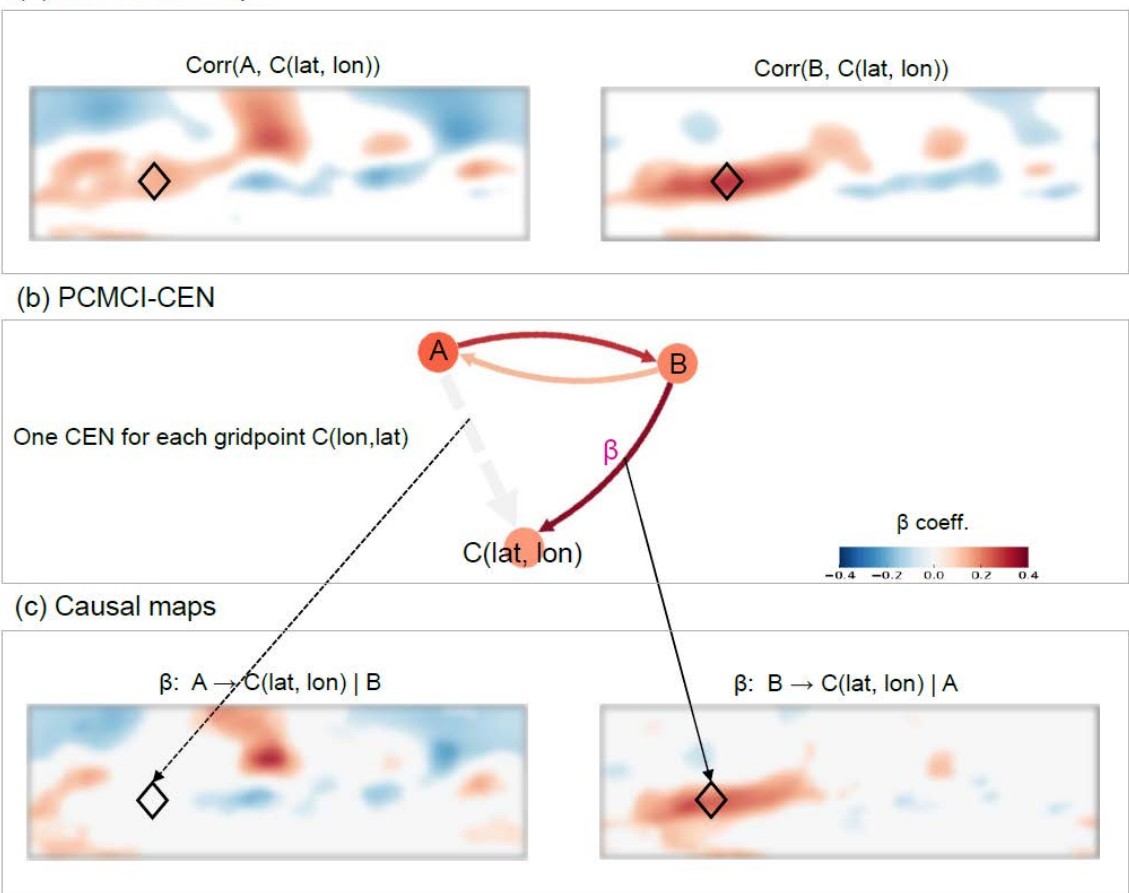

**Figure 1: Schematic explanation of causal maps.** Panel (a) shows the correlation maps for *A* with *C(lat,lon)* (left panel) and *B* with *C(lat,lon)* (right panel). Panel (b) shows an example of a CEN constructed with *A*, *B* and *C(lat,lon)* for a specific geographical position (identified with a diamond in the 2D maps). Panel (c) shows the corresponding causal maps showing the path coefficients *β* from *A* to *C*, conditioned on *B* and all autocorrelations (bottom-left panel) and from *B* to *C*, conditioned on *A* and all autocorrelations (bottom-right panel). The "|" denotes the conditioned-out actor: *A* for the right panel and *B* for the left panel. See text for further description.

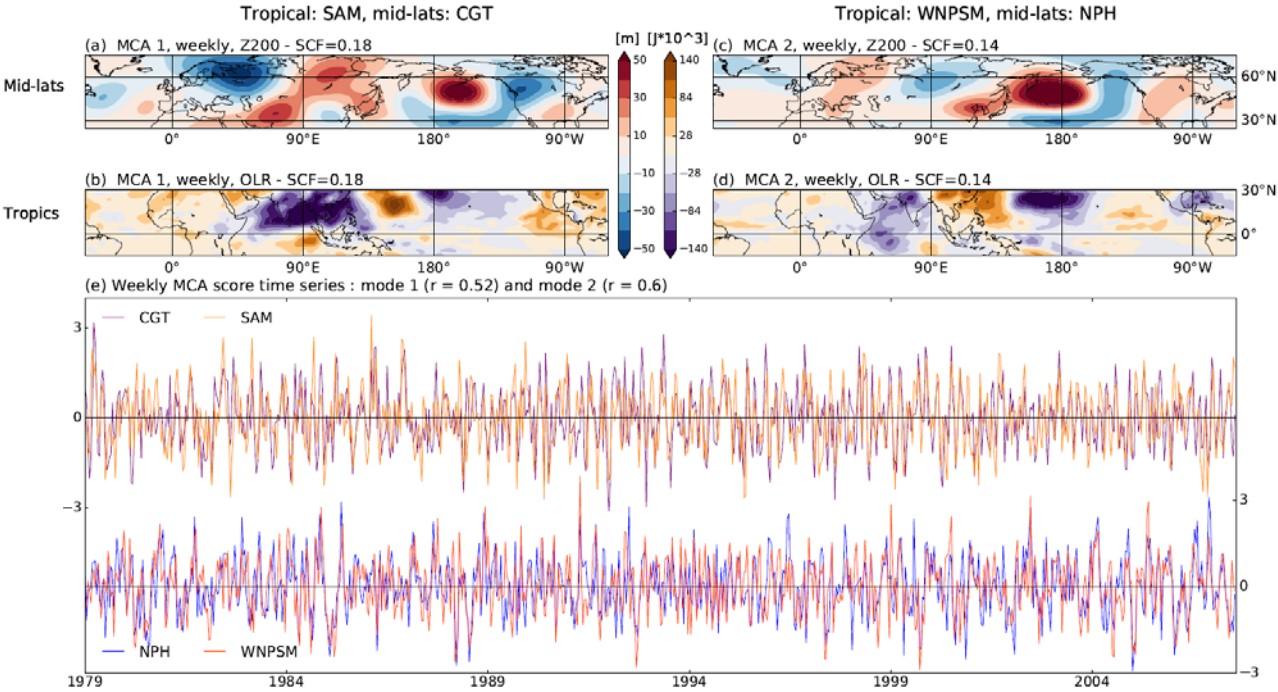

**Figure 2: MCA of mid-latitude Z200 and tropical OLR at intraseasonal timescales.** Panels (a) and (b) show the first MCA mode for mid-latitude Z200 (25-75° N) and tropical OLR (15°S-30°N), respectively, at the weekly timescale. The first MCA highlights the circumglobal teleconnection (CGT) pattern in the mid-latitudes and the South Asian monsoon (SAM) in the tropical belt. Panels (c) and (d): Same as for panel (a) and (b) but for the second MCA mode. This mode depicts the North Pacific High (NPH) in the mid-latitudes and the western North Pacific summer monsoon (WNPSM) in the tropical belt. The squared covariance fraction (SCF) of each MCA mode is given on top of the panels. Panel (e) shows the time series of MCA scores for the two MCA modes at weekly timescale. Each MCA pattern has its own time series, i.e. one for tropical OLR and one for mid-latitude Z200 (note that different *y*-axes are used).

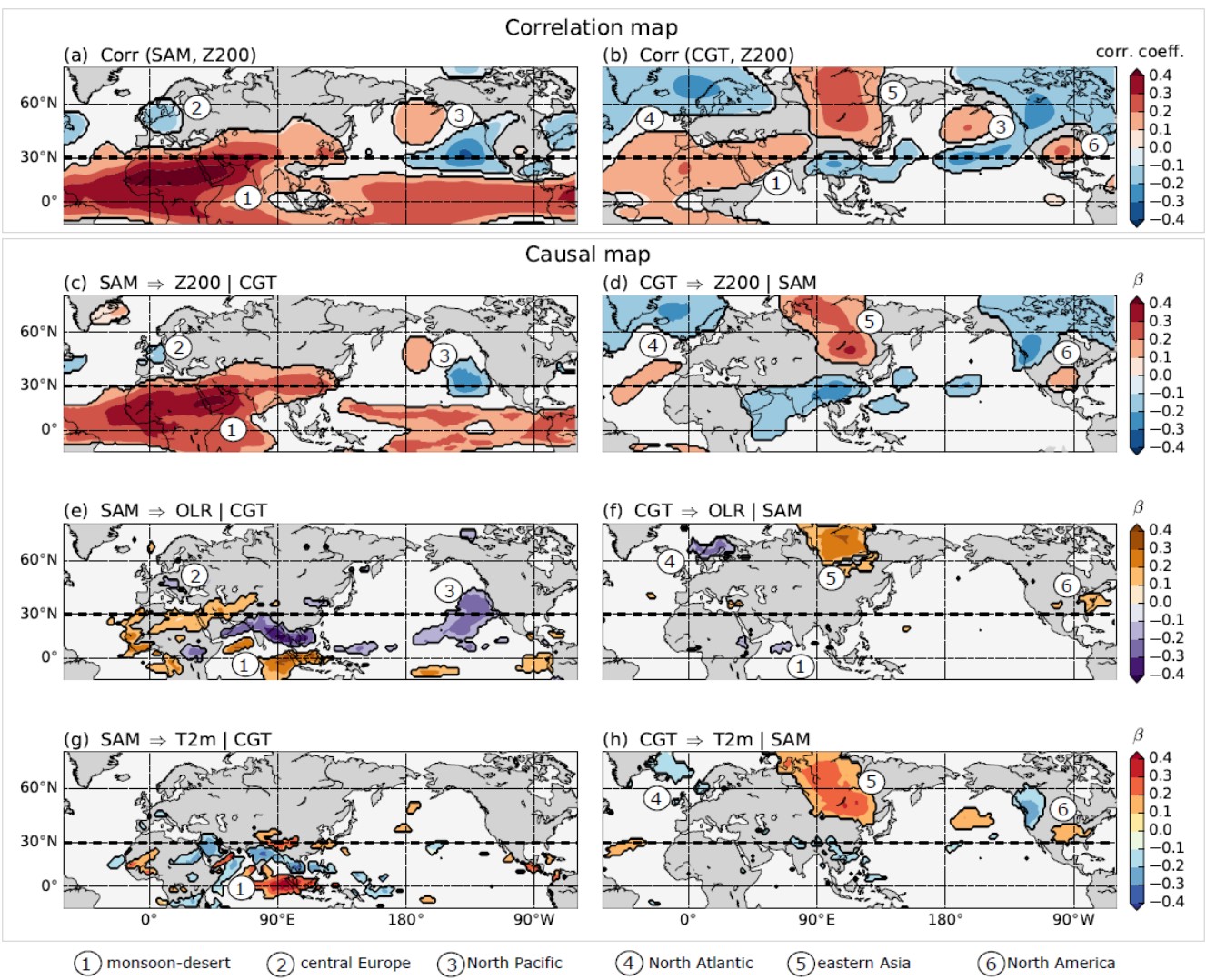

**Figure 3: Influence of MCA mode 1 on Northern Hemisphere circulation.** Panel (a): Correlation map between the weekly SAM time series and the Z200 field. Panel (b): Same as panel (a) but for the correlation between weekly CGT time series and the Z200 field. Panel (c): Path coefficient $\beta$ for link $SAM_{\tau=-1} \rightarrow Z200_{\tau=0}$ for a 3-actors CEN built with SAM, CGT and Z200. Here, the "|" denotes the conditioned-out actor: CGT. Panel (d): Same as panel (c) but for the link $CGT_{\tau=-1} \rightarrow Z200_{\tau=0}$. The "|" denotes the conditioned-out actor: SAM. Panels (e) and (g): Same as panel (c) but for the influence of SAM on OLR and T2m fields respectively. Panels (f) and (h): Same as panel (d) but for the influence of CGT on OLR and T2m fields respectively. Only path coefficients $\beta$ with $p < 0.05$ (accounting for the effect of serial correlations) and the robustness mask (see Fig. S8 in the Supplementary Material) are shown. The dashed black line located at 30°N shows the border between the tropical and the mid-latitude belt which separates OLR and Z200 analysis.

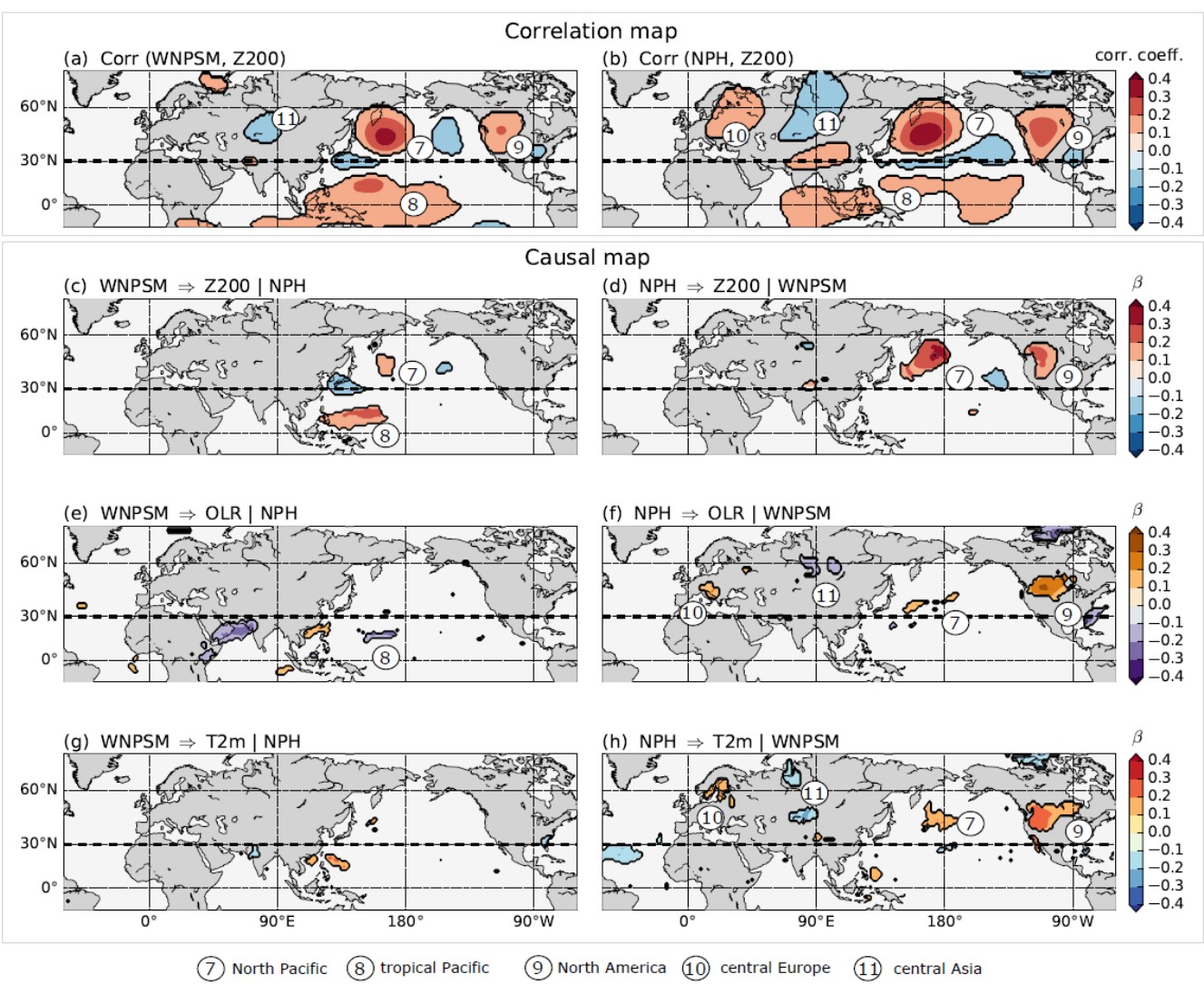

Figure 4: **Influence of MCA mode 2 on Northern Hemisphere circulation.** Panel (a): Correlation between the weekly WNPSM time series and the Z200 field. Panel (b): Same as panel (a) but for the correlation between weekly NPH time series and the Z200 field. Panel (c): Path coefficient $\beta$ for the link WNPSM $_{\tau=-1} \rightarrow$ Z200$_{\tau=0}$ in a 3-actors CEN built with WNPSM, NPH and Z200. Here, the "|" denotes the conditioned-out actor: NPH. Panel (d): Same as panel (c) but for the link NPH $_{\tau=-1} \rightarrow$ Z200$_{\tau=0}$. Here, the "|" denotes the conditioned-out actor: WNPSM. Panels (e) and (g): Same as panel (c) but for the influence of WNPSM on OLR and T2m fields respectively. Panels (f) and (h): Same as panel (d) but for the influence of NPH on OLR and T2m fields respectively. Only path coefficients $\beta$ with $p < 0.05$ (accounting for the effect of serial correlations) and the robustness mask (see Fig. S9 in the Supplementary Material) are shown. The dashed black line located at 30°N shows the border between the tropical and the mid-latitude belt which separates OLR and Z200 analysis.

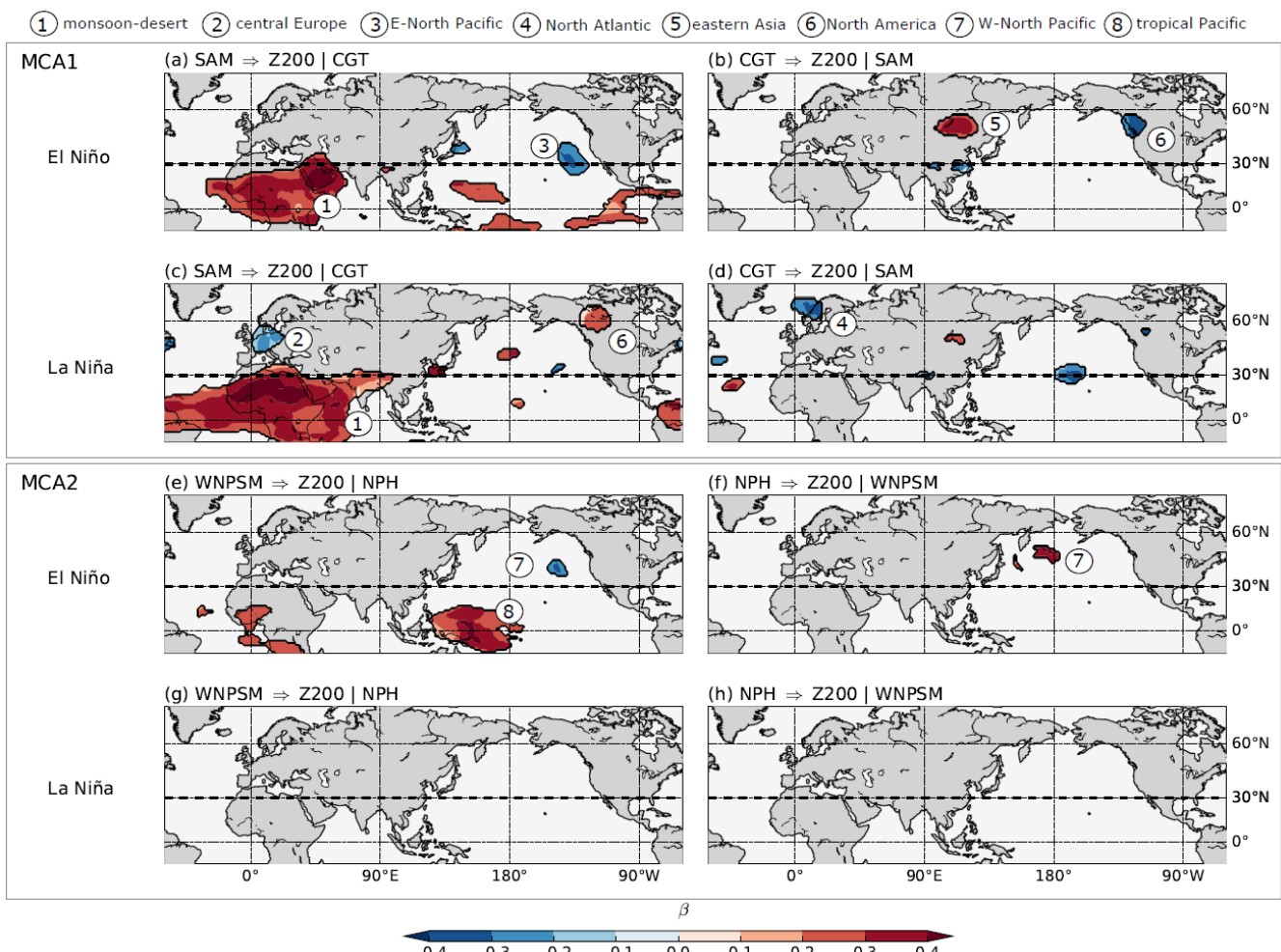

**Figure 5: Causal maps and ENSO influence.** Panel (a) shows the $\beta$ for link $\text{SAM}_{\tau=-1} \rightarrow \text{Z200}_{\tau=0}$ a 3-actors CEN built with SAM, CGT and Z200 during El Niño years. Here, the "|" denotes the conditioned-out actor: CGT. Panel (b): Same as panels (a) but for the link $\text{CGT}_{\tau=-1} \rightarrow \text{Z200}_{\tau=0}$. The "|" denotes the conditioned-out actor: SAM. Panel (c): Same as panel (a) but for La Niña years. Panel (d): Same as panel (c) but for the link $\text{CGT}_{\tau=-1} \rightarrow \text{Z200}_{\tau=0}$. Panel (e) and (g): Same as panels (a) and (c) but for the link $\text{WNPSM}_{\tau=-1} \rightarrow \text{Z200}_{\tau=0}$ from a 3-actors CEN built with WNPSM, NPH and Z200. Panel (f) and (h): Same as panels (e) and (g) but for the link $\text{NPH}_{\tau=-1} \rightarrow \text{Z200}_{\tau=0}$. Only path coefficients $\beta$ with $p < 0.05$ (accounting for the effect of serial correlations) and the robustness mask (see Fig. S12 in the Supplementary Material) are shown. The dashed black line located at 30°N shows the border between the tropical and the mid-latitude belt which separates OLR and Z200 analysis.

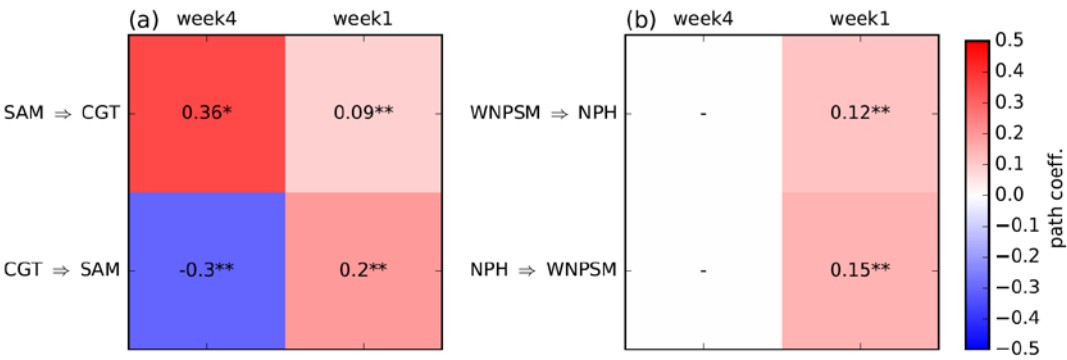

**Figure 6: Two-way causal link between tropical OLR and mid-latitude Z200.** Shown is the path coefficient for pairs of MCA time series. The CGT is studied along with the SAM, while the NPH is analysed together with the WNPSM. Panel (a) shows the path coefficient $\beta$ for the link $SAM_{\tau=-1} \rightarrow CGT_{\tau=0}$ over the 1979-2018 period (first row), and path coefficient $\beta$ for the link $CGT_{\tau=-1} \rightarrow SAM_{\tau=0}$ (second row). 4-weekly $\beta$ are shown in the left column, weekly $\beta$ values are shown in the right column. Panel (b): Same as for panel (a) but for $WNPSM_{\tau=-1} \rightarrow NPH_{\tau=0}$ and $NPH_{\tau=-1} \rightarrow WNPSM_{\tau=0}$ links respectively. $\beta$ values with $p < 0.1$ (0.05) are identified with one (two) asterisks.

**Acknowledgments**

This work has been financially supported by the German Federal Ministry for Education and Research of Germany (BMBF) via the Belmont Forum / JPI Climate project GOTHAM (grant no. 01LP1611A) and the BMBF Young Investigators Group CoSy-CC$^2$: Complex Systems Approaches to Understanding Causes and Consequences of Past, Present and Future Climate Change (grant no. 01LN1306A). The contribution of AGT was supported by NERC in the UK under the BITMAP project

(grant number NE/P006795/1) and in the STIMULATE project through the Weather and Climate Science for Service Partnership (WCSSP) India, a collaborative initiative between the Met Office, supported by the UK Government's Newton Fund, and the Indian Ministry of Earth Sciences (MoES).

**Data availability.** The data used in this article can be accessed by contacting the corresponding author.

**Author contributions.** GDC, DC, BvdH, JR and AGT designed the analysis. GDC performed the analysis and wrote the first draft of the paper. All authors contributed to the interpretation of the results and to the writing of the paper.

**Competing interests.** The authors declare that they have no conflict of interest.
