# Peer review of "Dominant patterns of interaction between the tropics and midlatitudes in boreal summer: Causal relationships and the role of timescales"

_Weather and Climate Dynamics, 2020_

## Referee Comment (RC1) · Anonymous Referee #1 · 25 May 2020

General comments: This manuscript purports to derive causal relationships between tropical convective activity and mid-latitude weather systems in the Northern Hemisphere. The approach is to apply maximum covariance analysis (MCA) between tropical convective activity (OLR) and mid-latitude geopotential height fields at 200hPa in order to identify the dominant modes of interaction – here the leading two MCA modes explaining roughly 30% of the variance. The potential causal interdependencies between the leading two MCA modes and other atmospheric fields, in this case 2m surface air temperature, are inferred via application of causal effect networks (CEN) at different time-scales and lags. The so-called causal maps reveal the unsurprising result that regions of enhanced tropical convection play a role in. modulating large-scale

synoptic variability within the midlatitude jets and notably in the regions associated with particular nodes of the circumglobal wave 5 pattern.

Specific comments: There is no literature review of other approaches to identifying causal relationships in climate data. One example is I. Horenko, S. Gerber, T.J. O'Kane, J.S. Risbey and D. Monselesan (2017) On inference and validation of causality relations in climate teleconnections, (In Nonlinear and Stochastic Climate Dynamics. Cambridge University Press, Eds. C. Franzke and T.J. O'Kane)

The initial application of MCA appears to perform a basic dimension reduction. The authors assert that "expert knowledge" is required in choosing the particular variables to calculate the cross covariances however there is no indication that any other combinations were examined. For example, OLR could be replaced with velocity potential – as in indices for the MJO – with similar results. The methodology applied here seems to be unable to answer if a sufficient set of covariates has been chosen apart. How, for example, do you test if the combination of actors is sufficient or even parsimonius? Can some form of information theoretic approach be applied for example Akaike or Bayesian?

Given the leading two modes of MCA appear to be in quadrature, how does MCA compare to EOF/PCA or even k-means? Apparently, many of the underlying assumptions are the same i.e stationarity etc It would help greatly if the authors could indicate if their approach is causal in the sense of Grainger given there appears to be no underlying stochastic model?

The analysis and attribution of the causal relationships is ultimately largely empirical, at times overly complicated and in some parts exceedingly verbose in description. The "causal maps" are very noisy and the reported relationships are very poorly represented from the patterns in the causal maps presented. It would greatly help the reader if the methodology was described in sufficient detail and better placed in context with other approaches, both in terms of dimension reduction and causal inference. This, in

combination with a more concise discussion of the physical properties of the modes would allow the reader to better judge the merits of the approach.

---

## Referee Comment (RC2) · Anonymous Referee #2 · 26 May 2020

General comments

The paper addresses relevant scientific questions of teleconnections between tropical convection related to monsoon activity and midlatitude circulation on subseasonal timescale, particularly authors focus on the boreal summer time. The paper presents a method for causal link visualization in the form of a 2D-map of regression coefficients describing linear relationship between the cause (monsoon activity and circulation patterns represented by maximum covariance analysis (MCA)) and response processes (2m air temperature and outgoing longwave radiation). The causal effect network (CEN) approach was tested previously on 1D reanalysis data in a number of

climate studies. Here the authors expanded the method to a grid-wise CEN analysis. They find that both-ways links exist in summer, which act on a 4-week timescale between the South Asian monsoon and the Northern Hemisphere circulation and on a 1-week timescale in the opposite direction. In addition, authors analyze causal links in the presence of different ENSO phases, however not in the mature state of ENSO phases but in summers before El-Ninos and La-Ninas reach a peak, which makes it somewhat difficult to compare with other studies that analyze monsoon-ENSO coupling mostly during the developed ENSO phases (e.g, Kawamura 1998,Kumar et al1999, Goswani and Xavier 2005, etc). This analysis suggests that the La Nina phase has a dominant effect on the summer link between the South Asian monsoon activity and the mid-latitude circulation as compared to the El Nino phase. Whereas the western North Pacific monsoon effect on the North Pacific High is stronger during the El Nino phase. In general, the scientific methods and assumptions are sufficiently described; however the methodology could still be improved by putting it into the context of studied teleconnections and for the traceability of the method and results. Also the result and discussion parts need further clarification. The language in general is fairly clear, however overloaded with abbreviations, which makes the paper somewhat difficult to follow.

Specific comments

1) Clarification on methodology: - Section 2.2: The choice of MCA is not clear as compared to other methods of dimension reduction. It would be helpful to describe what will happen with MCA modes after section 2.2. In Figure 2, the legend suggests four time series but one can only recognize two time series. - Section 2.4 is very generic; it would also be useful to know at some point what "A, B, C" are in the current analysis. Adding a table describing indexes and abbreviations separated in cause and response actors used for the causal effect analysis would be helpful. A discussion on the sensitivity of results to data-length would also be useful. - L219-223 and L310-315 should be in Methods because this text describes methodology and not the results.

2) Clarification on results and discussion: - L250-259: It is not clear what the purpose of this paragraph is. - L266-268: Mentioned patterns do not look "similar" at all to me. I would suggest to specify regions where similarities are seen by authors. - Explaining some of the results, authors interpret patches of beta-values on causal maps that look like noise. E.g., L280: "Although the CGT influence is mostly concentrated in the mid-latitude regions, one can see a negative causal effect of the CGT pattern on OLR values over the Bay of Bengal (Fig. 3f)." It looks like the effect that authors describe is a small dash over the Bay of Bengal, I cannot even see the color of the region, just the black contour color. Does the method behind causal maps take care of spatial noise? - L282: "Asia and North America are strongly affected by the CGT." It would be useful to support the qualitative judgment of the link-strength by providing beta-coefficient values in parentheses for this particular example and throughout the text, where link's strength from causal maps is described. - L455: "apparent paradox": I am not sure there is any paradox. Studies cited by the authors describe a trend in current observations and future climate change projections, which cover two different time periods, thus such comparison is not consistent. - L435-440: A comparison of teleconnections acting on subseasonal timescales from this study with those from other studies on interannual and decadal timescales is odd. - L56 and L496: A statement about paving the way to better predictions without further explanation is a bit bold. The CEN method has a potential to improve our understanding of climate processes but authors need to explain better how exactly this method can improve climate predictions.

3) Inaccurate region description: - L295: "Russia/Scandinavia": I would say "northern and eastern Europe" because this where non-zero beta values actually are. On the other hand, what does "non-corrected p values" from the caption mean, I do not find it explained. - L323: "over Kazakhstan" I would say "north of Kazakhstan" if the region enclosed by the contour is meant. Moreover, Kazakhstan is located north-east of the Caspian Sea not north-west of the Caspian Sea. - L319: "a few areas": Indeed these are three regions which can be named. - L412: "European Russia". I would rather say "northern and eastern Europe".

4) Figure 5: - During El Nino years, there is a link between SAM and Z200 in the tropical Pacific, which is not present during the La Nina years, therefore the concluding statement in the results, conclusions and abstract about strong effect of El-Nino only for the second MCA mode is confusing. - NPH and mode 2 results are not described in the text. - L417: "the pattern identified in Fig 5f with a low over central Europe and high over western Russia". I do not see a low-high dipole, the figure shows beta-coefficients not geopotential. - L419: "...wave-trains initiated by La Nina..." I do not follow this explanation. Figure 5f is about El Nino effects. Similarly, L456-458: "... if La Nina conditions would become...(Fig. 5f)". Figure 5f is about El Nino effects.

5) An extensive use of abbreviations makes the paper a bit difficult to follow. - Adding a table describing CEN actors abbreviations would be very helpful. - Abbreviation is introduced but never used in the manuscript such as EASM (L92) and SRP (L439). - BSISO abbreviation in L138 is not introduced.

---

## Short Comment (SC1) · 3 Jun 2020

SHORT COMMENT (combined for both Reviewer #1 and #2)

We thank the two anonymous reviewers for their valuable comments and constructive reviews. Both reports have been very helpful for us to identify those sections in the manuscript that require further improvements. Since the two reviews share several inputs and raise similar points, we provide here an initial short response addressing comments from both reviews. This is _not_ intended to be our final response, but a pitch of how we intend to address the reviews, and in the spirit of open discussion provided by Weather and Climate Dynamics, we hope to inspire a fruitful conversation with the reviewers to further improve our work. In our final response, we will describe in full detail how we plan to take all comments into account and implement them in the revised version of the paper. In particular:

- Both reviewers indicate that the causal maps appear "noisy" and that "a discussion on the sensitivity of results to data-length would also be useful". We have already done a range of sensitivity tests (see below) showing that the identified large-scale patterns are indeed robust. This step also makes the causal maps less noisy such that the robust patterns emerge better. This thus improves the visual appearance and interpretability of figures 3, 4 and 5.

- Further, we will improve on the design of the causal maps to make it easier for the reader to identify those regions in the maps that are described in the text

- We will improve the literature review and provide more context along with additional references to similar techniques previously applied to similar research questions

- We will improve the explanation of both the causal discovery algorithm and causal maps in the Methods section to help the readers better understanding each step

- We will highlight better why using causal discovery algorithms gives an advantage with respect to simple correlation techniques and why our work adds information on the tropical – mid-latitude interaction topic

- In general, we will work on the main text to improve the readability throughout the entire manuscript and make it is easier for the reader to follow the explanation of both the Methods and the Results sections

We have already implemented part of the suggested additional analyses, to verify the robustness and sensitivity of the results, and we provide some of the results below.

ROBUSTNESS OF CAUSAL MAPS

Both reviewers have pointed out that the causal maps appear noisy and that a test of robustness on how sensitive these results may be to different temporal periods is needed. This is an important point, and hence we addressed it immediately.

To address both issues, we have pursued the following strategy: we calculate causal maps several times (10x) by removing each time the 10% of the total time series length. Here, we have 40 years of data, we thus remove in each step 4 consecutive years. As an example, in the first iteration we remove years 1979-1982, in the second iteration years 1983-1986 and so on. As a result, we obtain

10 causal maps. Then, collapse the information to one single map (Fig. 1 in this document) showing (for each grid-point) the fraction of times in which a significant causal link has been found based on the reduced data sets. Figure 1 reported below shows the results of this process for MCA 1. Regions that show a dark purple colour correspond to those identified in all ten causal maps. We can then set a threshold, for example 0.7 (meaning that a certain region is identified in 70% of the times) and use this as a mask for our causal maps (See Fig. 2 in this document). This results in a less noisy plot, where the plotted regions also represent the robustness of the analysis.

[Figure]

Fig. 1: Robustness test for causal maps as shown in Fig. 3 of the main manuscript. Dark purple shows regions that show a significant causal link at alpha = 0.05 (after applying the false discovery rate correction) in all the ten causal maps obtained by iteratively removing a set of 4 consecutive years each time, with a value of 1 meaning that a certain regions is always identified.

[Figure]

Fig. 2: As for Fig. 3 in the main text but masking out all regions that appear less then 70% of the times (see Fig. 1 in this document).

COMPARISON WITH EOF PATTERNS

We address the comment n3 by reviewer #1 on how our MCA patterns compare to EOF patterns. In Fig. 3, we show the first 5 EOF patterns for both Z200 and OLR. We calculate the spatial correlation between EOF and MCA patterns. For Z200, MCA 1 shows the strongest correlation with EOF 2 (r ~ 0.8), unsurprisingly since this pattern represents the circumglobal teleconnection pattern, which has been previously shown to be linked to the second EOF of Z200 (Ding and Wang 2005, Di Capua et al. 2020). MCA 2 has a stronger spatial correlation (r ~ 0.6) with EOF 1. For OLR, MCA 1 shows the strongest correlation with EOF 2 (r ~ 0.5), while MCA 2 has the strongest correlation with EOF 5 (r ~0.4).

Thus, with the only exception of OLR MCA 2, all MCA patterns are closely related with the first two EOFs for both Z200 and OLR. This comparison is useful to show how important each pattern is when a dimension reduction analysis is applied on Z200 and OLR separately. Note that the amount of variance explained is relatively low, but this depends on the fact that the interannual variability has been removed, thus leaving only the disturbances from the year-specific mean state. However, since in our present work, we are interested in identifying those patterns that evolve _simultaneously_ (due to some dynamical coupling between the two fields), we applied MCA to identify those patterns that can explain shared covariance, which is an objective that cannot be addressed by using EOF analysis alone.

[Figure]

Fig. 3: EOF analysis. The right column shows the first five EOF patterns for Z200, the left column shows the first five EOF patterns for OLR. In the title of each panel, the spatial correlation values with the MCA patterns reported in Fig. 2 of the main manuscript are shown. Red font highlights those EOFs that exhibit the strongest overall correlation with the MCA patterns discussed in our manuscript.

**MCA WITH VERTICAL VELOCITY AND VELOCITY POTENTIAL**

Following the recommendations of reviewer #1, we provide a sensitivity test for the identified MCA patterns by substituting OLR with velocity potential and vertical velocity. Note that, we have originally applied MCA on mid-latitude Z200 and tropical OLR because we are interested in studying the relationship between mid-latitude circulation patterns and tropical convection. Thus, we are restricted to variables representing tropical convection when attempting to provide a comparable analysis.

We originally selected OLR both because it captures strong convective clouds (but in a more smoothed way than expected for direct rainfall estimates), and because OLR is also used, for example, to define the BSISO index that describes the essential evolution of convective activity over the Indian Ocean region. Figure 4 shows the first two MCA patterns for Z200 paired with vertical velocity, while Fig. 5 shows the same for Z200 paired with velocity potential.

As we are interested in a proxy for convective activity, we perform our MCA analysis using vertical velocity, another proxy of convection (as strong upward vertical motions occur along with strong convective activity). The MCA patterns obtained when pairing vertical velocity with Z200 show highly consistent results with respect to those found for Z200 and OLR (Fig. 2 in the main text), thus demonstrating the robustness of the original MCA results obtained with OLR. (Note that in Fig. 4 upward motion has a negative sign since vertical velocity is expressed in Pa/s).

When we use velocity potential (Fig. 5 in this document), the original MCA 1 pattern obtained using OLR is still well recovered (with a wave-5 pattern in Z200 and low velocity potential over the Indian summer monsoon region). The MCA 2 pattern however shows a less pronounced agreement, only

partly capturing the OLR pattern in the western Indian Ocean but failing to represent the WNPSM convective activity. A reason for this discrepancy is that velocity potential provides an even smoother proxy for divergence, which is very strong in the Indian monsoon region, and apparently less pronounced in relation to the WNPSM.

[Figure]

Fig. 4 MCA results for Z200 and velocity potential.

[Figure]

Fig. 5 MCA results for Z200 and vertical velocity.

---

## Author Comment (AC1) · 13 Jul 2020

Dear Referee #1,

we have finalized our response by addressing all suggestions arisen in this review. Please find our response in the pdf document in attach to this message.

Kind regards, G. Di Capua

Please also note the supplement to this comment:
https://wcd.copernicus.org/preprints/wcd-2020-14/wcd-2020-14-AC1-supplement.pdf

---

## Author Comment (AC2) · 13 Jul 2020

**Response to review #2**

We thank anonymous referee #2 for the constructive review and helpful comments that have greatly helped us to improve our work in the revised manuscript. The main improvements in the response to reviewer #2 are summarized as follows:

- We have added a robustness test to check the sensitivity of the detected causal links when the time period is changed
- We have improved the visualization of the causal maps by reducing the noise and adding labels to better identify each region when described in the text
- We have expanded the explanation in the Methods section
- We have checked the description of each region in the Results section

We have taken into account all suggestions made by the reviewer and a point-by-point response to each comment is reported below. Please note that in the following text the referee's comments are highlighted in bold font, while our answers are in regular font.

**Specific comments**

**1) Clarification on methodology: - Section 2.2: The choice of MCA is not clear as compared to other methods of dimension reduction.**

We choose MCA over other methods of dimension reduction because we are interested to identify those patterns that evolve simultaneously and may be causally related (via e.g. dynamical coupling between multiple variables). Thus, we applied MCA to identify those patterns that can explain shared covariance, which is an objective that cannot be addressed by using EOF analysis alone. We will explain this point explicitly in the revised manuscript: "Among the available correlation based methods to highlight strong co-variability and reduce the dimensionality of a spatiotemporal dataset, MCA allows identification of patterns in pair of variables that evolve simultaneously and may be causally related (via e.g. dynamical coupling between multiple climatological fields). MCA detects patterns that can explain shared covariance, which cannot be achieved using other dimensionality reduction methods that consider individual variables separately, such as empirical orthogonal function (EOF) analysis. However, for providing a complete picture we will also discuss the corresponding EOF patterns and the fraction of variance explained for comparison with our MCA results."

**It would be helpful to describe what will happen with MCA modes after section 2.2.**

We now explain in more detail what happens to times series identified by using MCA in section 2.2: "Here, we select the first two MCA modes representing the dominant patterns of co-variability between tropical convection and mid-latitude circulation, and calculate time series for each MCA mode. These time series will be used as input for the causal discovery algorithm (see sections 2.3 and 2.4)."

**In Figure 2, the legend suggests four time series but one can only recognize two time series.**

We agree that in the first version of Fig. 2 it was difficult to recognise two time series. We have changed the colours to represent the two pairs of time series and adopted a different aspect ratio for the axes to better show the four time series. See Fig. R1 in this document.

Figure R1. Revised version of Fig. 2.

**Section 2.4 is very generic; it would also be useful to know at some point what "A, B, C" are in the current analysis.**

We will include this suggestion in the revised manuscript. In the Results section, we will make explicit how the variables used compare to the examples given in the method section. For example: *"Referring to the schematic illustrated in Fig. 1 and following the PCMCI algorithm explanation (section 2.3), here A and B time series are represented by the SAM and CGT time series respectively, while C(lon, lat) is represented by Z200, OLR and T2m fields."* Moreover, we have also added a more detailed explanation on how these time series are used in the causal discovery algorithm (see reponse to reviewer #1, point 4).

**Adding a table describing indexes and abbreviations separated in cause and response actors used for the causal effect analysis would be helpful.**

Following the reviewer's suggestion we have added a table (Table R1 in this document) to better identify each time series/field used (see also point 5 in this response).

| ABBREVIATION | FULL NAME                                      | DIMENSIONS                          |
|--------------|------------------------------------------------|-------------------------------------|
| BSISO        | Boreal summer intraseasonal oscillation        | 1D time series                      |
| SAM          | South Asian monsoon - MCA mode 1 OLR           | 2D spatial pattern + 1D time series |
| CGT          | Circumglobal teleconnection pattern – MCA mode | 2D spatial pattern + 1D time series |
|              | 1 Z200                                         |                                     |
| WNPSM        | Western North Pacific summer monsoon – MCA     | 2D spatial pattern + 1D time series |
|              | mode 2 OLR                                     |                                     |
| NPH          | North Pacific High – MCA mode 2 Z200           | 2D spatial pattern + 1D time series |
| Z200         | Geopotential height at 200 hPa                 | 2D field + time                     |
| OLR          | Outgoing longwave radiation                    | 2D field + time                     |
| T2M          | 2m temperature                                 | 2D field + time                     |

**Table R1.**

**A discussion on the sensitivity of results to data-length would also be useful.**

We address this comment by providing a robustness test by repeated calculation of the causal maps and screening for robust regions in the final results. This step also makes the causal maps less noisy, such that robust patterns emerge better, improving the visual appearance and interpretability of Figures 3, 4 and 5 in the revised version of the manuscript. We describe in detail how this test is performed: *"Finally, to test the robustness of our causal maps to the choice of time period and to reduce non-robust small-scale features, we repeatedly calculate causal maps for reduced time series length. In 10 trials we removed a consecutive time record of ~10% (4 years) of the entire period. For ENSO dependent causal maps, we have shorter time series and we thus remove only one year in each trial, leaving a set of 14 causal maps for La Niña events and 13 causal maps for El Niño events. As a result, we obtain an ensemble of causal maps and apply the false discovery rate correction to their pvalues. Then, both for the 1979-2018 period and for El Niño and La Niña years separately, we masked out areas where less than 70% of the trials indicated a significant causal link. This gives an indication of robustness of our findings and suppresses noise." The masks obtained in this way and used to produce new Fig. 3,4 and 5 will be inserted in the revised supplementary material and are reported below (Figs. R2-R4 in this document).*

---

## Author Response (AR1)

Dear Editor,

Herewith we submit the revised version of our manuscript "Dominant patterns of interaction between the tropics and mid-latitudes in boreal summer: Causal relationships and the role of timescales" by Di Capua et al. to the Copernicus Journal *Weather and Climate Dynamics*.

We have addressed all comments and suggestions made by the two anonymous reviewers. The revised manuscript has improved both in clarity of the content and robustness of the analysis.

The revised version of the manuscript contains all changes as indicated in the two responses to the reviewers' comments published in the public discussion on WCDD. A point-by-point response to all reviewers' comments are found below, together with a "track changes" version of the manuscript.

We very much hope that *Weather and Climate Dynamics* will considers this manuscript for publication.

Yours sincerely,

Potsdam, 20.07.2020

Giorgia Di Capua on behalf of all authors

**Response to review #1**

We thank anonymous referee #1 for the constructive review and helpful comments that have greatly helped us to improve our work in the revised manuscript. The main improvement are summarized as follows:

- We have performed a series of sensitivity tests which show that our findings are robust, and which have reduced noise in some plots and thereby their visual appearance
- We have improved the description of the methodology section and the visualisation of the results
- We have added further background literature on the use of causality methods in atmospheric science

We have taken into account all suggestions made and a point-by-point response to each comment is reported below. Please note that in the following text the referee's comments are highlighted in bold font, while our answers are in regular font.

**Specific comments**

- There is no literature review of other approaches to identifying causal relationships in climate 1. data. One example is I. Horenko, S. Gerber, T.J. O'Kane, J.S. Risbey and D. Monselesan (2017) On inference and validation of causality relations in climate teleconnections, (In Nonlinear and Stochastic Climate Dynamics. Cambridge University Press, Eds. C. Franzke and T.J. O'Kane) We thank the anonymous reviewer for this suggestion. We included in our revised version of the manuscript a paragraph briefly describing other causal approaches applied to atmospheric sciences (lines 114-121): "In recent years, several approaches have been applied to identify causal relationships in climate and atmospheric sciences (Runge et al., 2019b), ranging from Granger causality (McGraw and Barnes, 2018, 2020; Samarasinghe et al., 2019) to causal (Bayesian) graphical models (Pearl, 2000, Ebert-Uphoff and Deng, 2012a, 2012b; Horenko et al., 2017) and conditional independence-based network discovery methods for time series (Runge et al., 2019a). These studies have shown the ability of causal discovery tools to improve our understanding of several atmospheric circulation interactions such as Arctic – mid-latitudes connections (McGraw and Barnes, 2020; Samarasinghe et al., 2019), synoptic-scale disturbances between boreal summer and boreal winter (Ebert-Uphoff and Deng, 2012a) and the relationship between ENSO and surface temperature in the American continent (McGraw and Barnes, 2018)."
- 2. The initial application of MCA appears to perform a basic dimension reduction. The authors assert that "expert knowledge" is required in choosing the particular variables to calculate the cross covariances however there is no indication that any other combinations were examined. For example, OLR could be replaced with velocity potential as in indices for the MJO with similar results.

We thank the anonymous reviewer for suggesting this interesting test and we have expanded our analysis by considering the results obtained when other variables are used. We applied MCA on midlatitude Z200 and tropical OLR because we are interested in studying the relationship between midlatitude circulation patterns and tropical convection. Thus, we are focussing on variables representing tropical convection when attempting to provide a comparable analysis. We originally selected OLR because it captures strong convective clouds (which is a smoother signal than direct rainfall estimates), and because OLR is also used, for example, to define the BSISO index that describes the essential evolution of convective activity over the Indian Ocean region. In the revised version of the manuscript, we will provide a series of sensitivity tests for the identified MCA patterns by substituting OLR with velocity potential or vertical velocity (a proxy of convection). Figure S5 (in the revised version of the Supplementary Material) shows the first two MCA patterns for mid-latitude Z200 paired with tropical vertical velocity (note that upward motion has a negative sign since vertical velocity is expressed in Pa/s), while Fig. S6 shows the same for Z200 paired with velocity potential. The MCA patterns obtained when pairing vertical velocity with Z200 (figure S5) show highly consistent results with respect to those found for Z200 and OLR (Fig. 2 in the main text), demonstrating the robustness of the original MCA results obtained with OLR. When we use velocity potential (Fig. S6), the MCA 1 pattern strongly resembles that originally obtained using OLR (with a wave-5 pattern in Z200 and low velocity potential over the Indian summer monsoon region). The MCA 2 pattern however shows less agreement: It correctly captures the OLR pattern in the western Indian Ocean but does not represent the WNPSM convective activity patterns. A reason for this discrepancy is that velocity potential provides a much smoother proxy for upper-level divergence than OLR, which is very strong in the Indian monsoon region, and apparently less pronounced in relation to the WNPSM. We briefly comment on this in the revised main text (lines 350-355): "We also investigate whether the obtained MCA patterns are sensitive to the choice of OLR in representing tropical convective activity is represented by enhanced upward motions, shows qualitatively the same patterns as those in Figs. 2b,d (see Fig. S5 in the Supplementary Material). When velocity potential is used instead of OLR, the first MCA mode still closely resembles the OLR/Z200 MCA mode 1, while the second MCA mode only partly captures features in the western Indian Ocean (see Fig. S6 in the Supplementary Material)."

**The methodology applied here seems to be unable to answer if a sufficient set of covariates has been chosen apart. How, for example, do you test if the combination of actors is sufficient or even parsimonius? Can some form of information theoretic approach be applied for example Akaike or Bayesian?**

We consider causal discovery here and not a prediction task of any of the actors, for which criteria such as those mentioned are indeed important. Hence, the choice of included actors is subject to the hypothesis underlying the analysis setup. One could, however, phrase causal discovery, as in Granger's work, as a prediction problem. On the other hand, a causal interpretation rests on a number of assumptions and we discuss limitations related to causal sufficiency and other assumptions made in the discussion in the revised manuscript (lines 644-659): "Finally, it should not be forgotten that in the context of the present work, causal interpretation rests upon several assumptions, such as the causal Markov condition, faithfulness, causal sufficiency, stationarity of the causal links and assumptions about the dependence-type (Runge, 2018). These assumptions can be violated in a real system and it is important to be aware of the associated typical challenges for causal discovery in Earth system sciences (Runge et al., 2019). Causal sufficiency requires that all relevant actors in a specific system are accounted for. Here, given the limited set of actors analysed, we cannot rule out that other excluded actors may act as important (common) drivers. Therefore, the obtained links can be considered causal only with respect to the specific set of actors used here. However, the absence of a link can still be interpreted as a likely indication that no direct physical connection among the respective variables exists. Moreover, we assume linear dependencies and stationarity for the detection of the causal links. While linearity has been shown to be a useful assumption in previous work (Di Capua et al., 2020), monsoon dynamics behaves partly nonlinearly and therefore, our causal networks only capture some part of the underlying mechanisms by construction. Also, the SAM teleconnections might well behave in an nonstationary manner on decadal time-scales (Di Capua et al., 2019; Robock et al., 2003). We therefore cannot rule out that (multi-)decadal oscillations such as the Pacific Decadal Oscillation may influence our results. However, the amount of reliable data is limited and this prohibits the application of nonlinear measures or study of effects of nonstationarity."

**3. Given the leading two modes of MCA appear to be in quadrature, how does MCA compare to EOF/PCA or even k-means?**

We thank the anonymous reviewer for raising this point. We have now performed a comparison between MCA patterns and EOF patterns. In Fig. S4 (in the revised version of the Supplementary Material), which will be included in the revised Supplementary Material, we show the first 5 EOF patterns for both Z200 and OLR. We calculate the spatial correlation between all EOF and MCA patterns. For Z200, MCA 1 shows the strongest correlation with EOF 2 (r ~ 0.8). This is consistent with previous literature showing that the circumglobal teleconnection pattern (as captured by Z200 of

MCA1), is linked to the second EOF of Z200 (Ding and Wang 2005, Di Capua et al. 2020). MCA 2 has a strong spatial correlation ( $r \sim 0.6$ ) with EOF 1. For OLR, MCA 1 shows the strongest correlation with EOF 2 ( $r \sim 0.5$ ), while MCA 2 has the strongest correlation with EOF 5 ( $r \sim 0.4$ ). Thus, with only the exception of OLR MCA 2, all MCA patterns are closely related to the first two EOFs for both Z200 and OLR. This comparison shows that the identified MCA patterns are also on a regional level important in explaining the variability. Note that the fraction of variance explained is relatively low (for all EOFs), but this relates to the prior removal of interannual variability, thus leaving only the disturbances from the year-specific mean state. In our present work, we are interested in identifying those patterns that evolve simultaneously (due to the dynamical coupling between the two fields), and therefore we applied MCA to identify those patterns that can explain *shared* covariance, which is not captured by separate EOF analyses. We briefly comment on this in the main text (lines 342-349): "We compare the patterns obtained with MCA with those obtained with EOF analysis of Z200 and OLR fields (see Fig. S4 in the Supplementary Material). We find that the closest match of the Z200 MCA mode 1 pattern is with Z200 EOF 2 (spatial correlation ~ 0.8), while the closest match of Z200 MCA mode 2 is with EOF 1 (spatial correlation ~ 0.6). OLR MCA mode 1 has the closest match with EOF 2 (spatial correlation  $\sim 0.5$ ), while OLR MCA mode 2 has the closest match with EOF 5 (spatial correlation ~ 0.4). Thus, in general our MCA patterns also reflect the first two EOFs of Z200 and OLR indicating that they explain an important fraction of the regional variability. Nevertheless, here we are interested in those patterns that can explain shared covariance, which cannot be achieved by using EOF analysis alone. Therefore, we use the MCA-defined patterns for the following part of the analysis.".

**Apparently, many of the underlying assumptions are the same i.e stationarity etc It would help greatly if the authors could indicate if their approach is causal in the sense of Grainger given there appears to be no underlying stochastic model?**

Our definition of causal graphs follows Pearl's causal Bayesian networks (Pearl 2000) and our approach to estimate these graphs from data comes from the constraint-based causal discovery framework (Spirtes 2000), here adapted to time series (Runge et al. 2019). In the constraint-based causal discovery framework, the existence (or absence) of causal relations is based on conditional independencies among subsets of the lagged variables together with a number of assumptions (as listed in our Discussion section). If Granger causality is only applied to pairs of variables, Granger causality does not account for common drivers or indirect links as is the case in our framework. Further, the constraint-based causal discovery framework *in general* goes beyond Granger causality since it can also account for contemporaneous causal links. Here we only focus on lagged links. If Granger causality is meant in a full multivariate setting, our approach is asymptotically equivalent to Granger causality, but for finite samples Granger causality has much lower detection power since it does not deal well with the curse of dimensionality as investigated in detail in Runge et al. (2019).

**4. The analysis and attribution of the causal relationships is ultimately largely empirical, at times overly complicated and in some parts exceedingly verbose in description. The "causal maps" are very noisy and the reported relationships are very poorly represented from the patterns in the causal maps presented.**

We have taken the issue of noisiness raised by the anonymous reviewer very seriously, and combining this suggestion with the corresponding comment by anonymous reviewer #2, we have designed a robustness test that has removed much of the noise in the causal maps, greatly improving their visual appearance and interpretation. As a result, some of the more scattered regions that were described in the first version of the paper are now removed, and we can purely focus our description on the main, robust patterns. We describe this robustness in the revised manuscript (lines 295-303): "Finally, to test the robustness of our causal maps to the choice of time period, we calculate causal maps for a range of sub-periods. In 10 trials we removed 10% of the record (4 years). For ENSO-phase dependent causal maps, we have shorter time series and we thus remove one year in each trial, leaving a set of 14 causal maps for La Niña events and 13 causal maps for El Niño events. As a result, we obtain an

ensemble of causal maps and apply the false discovery rate correction to p-values of each single map. Then, both for the full period (1979-2018) and for El Niño and La Niña years separately, we masked out areas where less than 70% of the trials indicated a significant causal link, giving an indication of the robustness of our findings and at the same time suppressing noise."

This results in reduced noise in the new causal maps (see Fig. 3-5 in the revised version of the manuscript).

5. It would greatly help the reader if the methodology was described in sufficient detail and better placed in context with other approaches, both in terms of dimension reduction and causal inference. This, in combination with a more concise discussion of the physical properties of the modes would allow the reader to better judge the merits of the approach.

In the revised manuscript, we have improved the methodology section by adding a concrete example showing how the PCMCI algorithm works (also following the comments by the second reviewer, see point 1 in our response to reviewer #2), lines 213-257:

[revised manuscript text omitted]

**Response to review #2**

We thank anonymous referee #2 for the constructive review and helpful comments that have greatly helped us to improve our work in the revised manuscript. The main improvements in the response to reviewer #2 are summarized as follows:

- We have added a robustness test to check the sensitivity of the detected causal links when the time period is changed
- We have improved the visualization of the causal maps by reducing the noise and adding labels to better identify each region when described in the text
- We have expanded the explanation in the Methods section
- We have checked the description of each region in the Results section

We have taken into account all suggestions made by the reviewer and a point-by-point response to each comment is reported below. Please note that in the following text the referee's comments are highlighted in bold font, while our answers are in regular font.

**Specific comments**

**1) Clarification on methodology: - Section 2.2: The choice of MCA is not clear as compared to other methods of dimension reduction.**

We choose MCA over other methods of dimension reduction because we are interested to identify those patterns that evolve simultaneously and may be causally related (via e.g. dynamical coupling between multiple variables). Thus, we applied MCA to identify those patterns that can explain shared covariance, which is an objective that cannot be addressed by using EOF analysis alone. We explain this point explicitly in the revised manuscript (lines 169-174): "Among the available correlation based methods to highlight strong co-variability and reduce the dimensionality of a spatiotemporal dataset, MCA allows identification of patterns in pair of variables that evolve simultaneously and may be causally related (via e.g. dynamical coupling between multiple climatological fields). MCA detects patterns that can explain shared covariance, which cannot be achieved using other dimensionality reduction methods that consider individual variables separately, such as empirical orthogonal function (EOF) analysis. However, for providing a complete picture we will also discuss the corresponding EOF patterns and the fraction of variance explained for comparison with our MCA results."

**It would be helpful to describe what will happen with MCA modes after section 2.2.**

We now explain in more detail what happens to times series identified by using MCA in section 2.2 (lines 188-190): "Here, we select the first two MCA modes representing the dominant patterns of covariability between tropical convection and mid-latitude circulation, and calculate time series for each MCA mode. These time series will be used as input for the causal discovery algorithm (see sections 2.3 and 2.4)."

**In Figure 2, the legend suggests four time series but one can only recognize two time series.**

We agree that in the first version of Fig. 2 it was difficult to recognise two time series. We have changed the colours to represent the two pairs of time series and adopted a different aspect ratio for the axes to better show the four time series. See Fig. 2 in the revised version of the document document.

**Section 2.4 is very generic; it would also be useful to know at some point what "A, B, C" are in the current analysis.**

We will include this suggestion in the revised manuscript. In the Results section, we will make explicit how the variables used compare to the examples given in the method section. For example lines 379-381: "*Referring to the schematic illustrated in Fig. 1 and following the PCMCI algorithm explanation (section 2.3), here A and B time series are represented by the SAM and CGT time series respectively,*

*while C(lon, lat) is represented by Z200, OLR and T2m fields.*" Moreover, we have also added a more detailed explanation on how these time series are used in the causal discovery algorithm (see response to reviewer #1, point 4).

**Adding a table describing indexes and abbreviations separated in cause and response actors used for the causal effect analysis would be helpful.**

Following the reviewer's suggestion we have added a table (Table 1 in this in the revised version of the manuscript) to better identify each time series/field used (see also point 5 in this response).

**A discussion on the sensitivity of results to data-length would also be useful.**

We address this comment by providing a robustness test by repeated calculation of the causal maps and screening for robust regions in the final results. This step also makes the causal maps less noisy, such that robust patterns emerge better, improving the visual appearance and interpretability of Figures 3, 4 and 5 in the revised version of the manuscript. We describe in detail how this test is performed (lines 295-303): "Finally, to test the robustness of our causal maps to the choice of time period and to reduce non-robust small-scale features, we repeatedly calculate causal maps for reduced time series length. In 10 trials we removed a consecutive time record of ~10% (4 years) of the entire period. For ENSO dependent causal maps, we have shorter time series and we thus remove only one year in each trial, leaving a set of 14 causal maps for La Niña events and 13 causal maps for El Niño events. As a result, we obtain an ensemble of causal maps and apply the false discovery rate correction to their p-values. Then, both for the 1979-2018 period and for El Niño and La Niña years separately, we masked out areas where less than 70% of the trials indicated a significant causal link. This gives an indication of robustness of our findings and suppresses noise." The masks obtained in this way and used to produce new Fig. 3,4 and 5 are shown in the revised Supplementary Material (Figs. S8,S9 and S12).

**L219-223 and L310-315 should be in Methods because this text describes methodology and not the results.**

We have moved those lines "To extract the dominant co-variability patterns reflecting interactions between mid-latitude circulation in the Northern Hemisphere and tropical convection at intraseasonal time-scales, we follow Ding et al. (2011) and apply maximum covariance analysis (MCA) to OLR fields (used as a proxy for convective activity) in the tropical belt (15°S-30°N, 0°-360°E) paired with Z200 fields in the northern mid-latitudes (25°N-75°N, 0°-360°E)." (now lines 162-166) and "Here, we will derive causal maps using the time series obtained with MCA for modes 1 and 2 and Z200, OLR and T2m fields both for the entire time period (1979-2018) and for two subsets depicting different ENSO phases, to assess how the ENSO background state influences the causal relationships. El Niño (La Niña) summers are defined as summers preceding the El Niño (La Niña) peak in boreal winter. We thus obtain 14 La Niña years and 13 El Niño years (see Table 1 in the Supplementary material for a list of corresponding years and Fig. S1 for the associated SST anomaly composites). Although the strongest SST anomalies related to the ENSO phase are found in winter, warm (cold) SST patterns related to El Niño (La Niña) phases are already clearly developed during the preceding summers." (now lines 288-295) to the Methods section as suggested.

**2) Clarification on results and discussion: L250-259: It is not clear what the purpose of this paragraph is.**

We agree with anonymous reviewer #2 that a detailed description of BSISO can distract the reader from the main story line. We have moved this explanation into the SI and now refer to it only briefly in the main text (lines 322-325): "Using OLR composites, we explicitly show that the temporal evolution of the SAM convective activity at weekly time-scales resembles the evolution of the Boreal Summer Intraseasonal Oscillation (BSISO) (Goswami and Ajaya Mohan, 2001; Saha et al., 2012) (see Fig. S5-S6 and further discussion in the Supplementary Material)."

L266-268: Mentioned patterns do not look "similar" at all to me. I would suggest to specify

**regions where similarities are seen by authors.**

We thank the anonymous reviewer for pointing out that it was difficult to recognize in the figures the regions that we are interpreting in the text. We have addressed this comment by adding labels that are referred to in the main text, including the Results section. See new Figs. 3,4 and 5 in this document.

**Explaining some of the results, authors interpret patches of beta-values on causal maps that look like noise. E.g., L280: "Although the CGT influence is mostly concentrated in the mid-latitude regions, one can see a negative causal effect of the CGT pattern on OLR values over the Bay of Bengal (Fig. 3f)." It looks like the effect that authors describe is a small dash over the Bay of Bengal, I cannot even see the color of the region, just the black contour color. Does the method behind causal maps take care of spatial noise?**

We have taken the issue of robustness and potential noise in our causal maps seriously (see also our earlier reply and Figs. S8,S9 and S12 in the revised version of the Supplementary Material). The new figures 3, 4 and 5 (in the revised version of the manuscript) are now all produced using the robustness test described above (see our response to comment #1). As a result, the specific region described on line 280 (original manuscript) is now indeed masked out and we have updated the text correspondingly (lines 400-403): "*The CGT influence is mostly concentrated in the mid-latitude regions, and a significant and consistent negative causal effect of the CGT pattern on OLR values in the tropical regions can only be seen in a small area in the western Indian Ocean (Fig. 3f).*" In general, using the robustness test described above noisy patterns have been removed, enabling us to only discuss the main, large-scale patterns of interest.

**L282: "Asia and North America are strongly affected by the CGT." It would be useful to support the qualitative judgment of the link-strength by providing beta-coefficient values in parentheses for this particular example and throughout the text, where link's strength from causal maps is described.**

We thank the anonymous reviewer for this useful suggestion. We will add the values of the beta coefficients throughout the revised text to help the reader in the interpretation of the results.

**L455: "apparent paradox": I am not sure there is any paradox. Studies cited by the authors describe a trend in current observations and future climate change projections, which cover two different time periods, thus such comparison is not consistent.**

We have removed the sentence referring to the apparent paradox and rephrased the paragraph to make our point more carefully. The revised paragraph now reads (lines 620-626): "Future projections describe an increase in monsoon precipitation associated with increasing global mean temperature and thermodynamic arguments (Menon et al., 2013; Turner and Annamalai, 2012). Quantifying teleconnections between the tropics and mid-latitudes is important in order to better understand and constrain future changes in boreal summer circulation, as uncertainty may arise due to changing connections to remote regions. While simulations show great uncertainty in the ENSO response to global warming (Cai et al., 2015; Chen et al., 2017a, 2015, 2017b), observations show a La Niña-like warming trend in central-western Pacific SST (Kohyama et al., 2017; Mujumdar et al., 2012). "

**L435-440: A comparison of teleconnections acting on subseasonal timescales from this study with those from other studies on interannual and decadal timescales is odd.**

By comparing interannual and intraseasonal studies, we do not intend to imply that a similarity in the results obtained at different time scales *should* be expected. Nevertheless, a similarity in the pattern *is found* and this represents an outcome of our analysis that we believe needs to be discussed. In the discussion, we elaborate on what possible explanations for these findings there may be. Moreover, the similarity in these patterns between various time scales strongly suggests that there are interactions between the time scales – see for example the arguments of Sperber et al. (2000) who found a common mode of variability on intraseasonal and interannual time scales. Such commonality of patterns is necessary in order for the large scale forcing to be able to perturb the PDF at shorter time scales. See: Sperber et al. (2000) "Predictability and the relationship between subseasonal and interannual variability during the Asian summer monsoon", Quarterly Journal of the Royal Meteorological Society, 126: 2545-2574.

L56 and L496: A statement about paving the way to better predictions without further explanation is a bit bold. The CEN method has a potential to improve our understanding of climate processes but authors need to explain better how exactly this method can improve climate predictions.

We have added more in depth information on how CEN may help improving seasonal forecast in the revised version of the manuscript (lines 330-339): "A better understating of these teleconnections in observation can help to improve S2S forecasts. Verifying the existence and strength of causal teleconnections in forecast models, could help diagnose the origin of model biases. E.g. one could disentangle whether lower forecast skill (such as in the mid-latitude regions in summer) is related to local processes or to a misrepresentation of remote drivers. Beverley et al. (2019) showed that the CGT representation in seasonal forecasts is too weak. The CGT is important for predictability of summer extremes and its relationship with the SAM may provide some information to improve predictability. Therefore, these methods could help answering the question "where do model biases come from?" and help developing a physics-based bias correction. At the same time, CEN provide an encoded predictive model, which can be used for actual forecasting (Di Capua et al., 2019; Kretschmer et al., 2017; Lehmann et al., 2020)."

**3) Inaccurate region description: L295: "Russia/Scandinavia": I would say "northern and eastern Europe" because this where non-zero beta values actually are. On the other hand, what does "non-corrected p values" from the caption mean, I do not find it explained.**

We now show only p-values that are corrected using the false discovery rate correction, to reduce noise and non-robust results. We have also carefully checked the description of each region in the Result section.

**L323: "over Kazakhstan" I would say "north of Kazakhstan" if the region enclosed by the contour is meant. Moreover, Kazakhstan is located north-east of the Caspian Sea not north-west of the Caspian Sea.**

This region did not pass the new robustness test and was removed.

**L319: "a few areas": Indeed these are three regions which can be named.**

We have added regional labels in the causal maps and use those references in the text.

**L412: "European Russia". I would rather say "northern and eastern Europe".**

We will implement this suggestion in the revised manuscript.

**4) Figure 5: During El Nino years, there is a link between SAM and Z200 in the tropical Pacific, which is not present during the La Nina years, therefore the concluding statement in the results, conclusions and abstract about strong effect of El-Nino only for the second MCA mode is confusing.**

We thank the anonymous reviewer for pointing out this discrepancy. We now mention that both phases of ENSO affect the relationship between SAM and Z200 (lines 490-493): "*Thus, the second MCA mode (the WNPSM-NPH pair) has its strongest effect during El Niño summers, whereas the first MCA mode (SAM-CGT pair) is important during both La Niña and El Niño summers but with different characteristics*" and "*Nevertheless, during La Niña summers, the effect of the SAM-CGT mode is reinforced over Europe, North Africa and the Indian subcontinent and reaches northward towards Canada while during El Niño summers the effect of the SAM is mainly confined to the tropical belt. For the WNPSM-NPH pattern, a clear asymmetry between El Niño and La Niña summers is shown, with a stronger signal during El Niño (Fig. 5e,f) that is absent during La Niña years.*" (lines 596-600).

**NPH and mode 2 results are not described in the text.**

We now describe the results related to Mode 2 (lines 485-490): "In the western North Pacific, the most notable feature is the presence of both the WNPSM and NPH on the North Pacific only during El Niño

summers (Figs. 5e,f). During those summers, the positive causal effect of the WNPSM over the western North Pacific (Region 1 and 2 in Fig. 5e) intensifies in magnitude (absolute beta ~ 0.3-0.4) relative to the 1979-2018 mean pattern (Fig. 4c), although the geographical extent of Region 1 shrinks. Over the western tropical Pacific, in correspondence with the La Niña warm pool, a region of positive causal effect is shown (Region 2 in Fig. 5e). These features disappear during La Niña summers."

L417: "the pattern identified in Fig 5f with a low over central Europe and high over western Russia". I do not see a low-high dipole, the figure shows beta coefficients not geopotential. We have removed this sentence as this statement in not supported by the stricter robustness test applied in the new causal maps.

**L419: ": : :wave-trains initiated by La Nina: : :" I do not follow this explanation. We have removed this sentence as this statement in not supported by the stricter robustness test applied in the new causal maps.**

**Figure 5f is about El Nino effects. Similarly, L456-458: ": : : if La Nina conditions would become: : :(Fig. 5f)". Figure 5f is about El Nino effects.**

This mistake had been corrected by including the correct panel for Fig. 5c.

**5) An extensive use of abbreviations makes the paper a bit difficult to follow. – Adding a table describing CEN actors abbreviations would be very helpful. - Abbreviation is introduced but never used in the manuscript such as EASM (L92) and SRP (L439). - BSISO abbreviation in L138 is not introduced.**

Following the suggestion of the anonymous reviewer, we have added a table showing the full name of each abbreviation used throughout the manuscript and, when useful, its dimensions. We have removed abbreviations for EASM and SRP since they are not used later in the text. We now introduce the term BSISO both at its first appearance and in the abbreviation table.

**Dominant patterns of interaction between the tropics and midlatitudes in boreal summer: Causal relationships and the role of timescaletimescales**

10

Giorgia Di Capua1,2, Jakob Runge3, Reik V. Donner1,4, Bart van den Hurk2,5, Andrew G. Turner6,7, Ramesh Vellore8, Raghavan Krishnan8, and Dim Coumou1,2

15

- 1Potsdam Institute for Climate Impact Research, Potsdam, Germany

[revised manuscript text omitted]

- 215 { AII=-1, BII=-1, C(*lat, lonlat, lon)II=-1, ..., AII=-tmax, BII=-tmax, C(lat, lonlat, lon)II=-tmax*}. In this analysis, A and B represent the two MCA scores obtained for a selected MCA mode, while C(*lat, lon*) represents the grid point time series of a 2D field, e.g. T2m or Z200. In its first step, PCMCI iterates through partial correlations with increasing cardinality of conditions to remove the influence of common drivers and indirect links and estimate a preliminary set of parents. The first iteration of PC (cardinality 0) calculates the correlation between a selected time series, e.g. AT=0, and the past of any other available time series, { AT=-1, 200
  220 BT=-1, C(*lat, lon*)T=-1, ..., AT=-tmax, BT=-tmax, C(*lat, lon*)T=-tmax}, including its own past AT=-1, ..., For illustration purposes, we
- here provide an example for C(lat, lon), where  $\rho$  denotes the correlation and  $\tau$  is the lag that is being used in the network (in this example,  $\tau_{max} = -2$ ):

 $\begin{array}{c} \rho(C(lon, lat)_{\tau=o}, A_{\tau=-1}) = 0.32, p = 0.01 \ (5) \\ \rho(C(lon, lat)_{\tau=o}, A_{\tau=-2}) = 0.13, p = 0.1 \\ \rho(C(lon, lat)_{\tau=o}, B_{\tau=-1}) = 0.35, p = 0.005 \\ \rho(C(lon, lat)_{\tau=o}, B_{\tau=-2}) = 0.23, p = 0.058 \\ \rho(C(lon, lat)_{\tau=o}, C(lon, lat)_{\tau=-1}) = 0.41, p = 0.01 \\ \rho(C(lon, lat)_{\tau=o}, C(lon, lat)_{\tau=-2}) = -0.16, p = 0.06 \\ \hline \text{Applying a significance level } \alpha = 0.05, \text{ only three actors are significantly correlated with } C(lat, lon) \text{ at the chosen time lag.} \\ \hline \text{These form the initial preliminary set of parents for } C(lat, lon)_{\tau=-1}, B_{\tau=-1}, A_{\tau=-1} \end{array}$

Next, partial correlations between C(lat, lon) and each actor in  $P_{C(lon, lat)}^{0}$  are calculated by conditioning on the strongest preliminary parent:

$$p(C(lat, lon)_{\tau=0}, C(lat, lon)_{\tau=0}, P_{\tau=1}|B_{\tau=-1}| = 0.35, p = 0.02$$
(7)

$$p(C(lat, lon)_{\tau=0}, P_{t=-1}|C(lat, lon)_{\tau=-1}) = 0.25, p = 0.03$$

[revised manuscript text omitted]

**(a) Correlation maps**

---

## Author Response (AR2)

**Response to the editor**

We thank the anonymous reviewers and the editor for the positive response regarding our manuscript "Dominant patterns of interaction between the tropics and mid-latitudes in boreal summer: Causal relationships and the role of timescales" (wcd-2020-14).

We have revised the manuscript following all minor suggestions made by the second reviewer in his/her thorough review. Our point-by-point response is found below.

We thank the reviewers for finding the time to carefully and effectively reviewing the manuscript and the editor for her supportive role in the reviewing process.

**Point-by-point response to Reviewer #2**

1) L51, L279, L286: Change "the causal maps that plot" to "the causal maps show"
   *We have changed the manuscript accordingly to the reviewer's suggestion.*

   2) L57-58: "and therefore works towards improved sub-seasonal and climate projections". I think the authors wanted to say "sub-seasonal predictions and climate projections" as predictions and projections are two different types of climate simulations: the former one is the initial-value or initial and boundary value problem whereas the latter is the boundary-value problem. On the other hand, the sentence could be finished before "and therefore".
   *We thank the anonymous reviewer for point out this missing word. We have changed the sentence to "This study paves the way for process-based validation of boreal summer teleconnections in (sub-)seasonal forecast models and climate models and therefore works towards improved sub-seasonal predictions and climate projections. " (lines 56-58).*

   3) L247-248: "By repeating this step for each variable…" But also repeating this step for all lat and lon, isn't it? Or from which step the analysis should be repeated for each lon and lat, please specify this more clearly.
   *We have clarified this sentence: "By repeating this step for each variable (and for each longitude and latitude position), preliminary sets of parents are estimated." (lines 247).*

   4) L262: Eta should be also explained for completeness of the description.

   *We have added the definition of eta in the manuscript:*
   *Finally, we estimate the Causal Effect Network (CEN) (Kretschmer et al., 2016; Runge et al., 2015a) among A, B and C(lat,lon) by applying standardized multiple regression of each actor onto its causal parents identified via PCMCI, i.e., for Y ∈ in $A_t$, $B_t$, $C(lat,lon)_t$ and the parents P:*

   $$Y_t = \sum_i \beta_i X_i + \eta_Y \qquad (13)$$

   *where $X_i \in P\{Y\}$, i = 1, .., N, i.e. the set of N parents of $Y_t$ and $\eta_Y$ is the residual of $Y_t$ (i.e. the difference between the observed value $Y_t$ and the value obtained by the linear regression on the causal parents $\sum_i \beta_i X_i$).*

   5) L310: Should be "Figure 2a-d show"
   *We have corrected this expression following the reviewers suggestion (line 312).*

6) L536-541 is one sentence, which is too long and difficult to follow.
*We have spitted the sentence in two to improve its readability: "Two pairs of co-varying patterns are identified: a) convective activity of the South Asian monsoon (SAM) paired with a mid-latitude wavenumber-5 wave train resembling the circumglobal teleconnection (CGT) pattern and b) convective activity of the western North Pacific summer monsoon (WNPSM) paired with a second wave-5 circumglobal wave pattern with its strongest action centre represented by the North Pacific High (NPH). This second mid-latitude wave pattern is phase shifted with respect to the CGT pattern, to the longitudinal position of WNPSM monsoonal convection in the tropics." (lines 537-543).*

7) L546: " ... dominate" what?
*We have changed the sentence to "At longer timescales (from monthly to seasonal) slowly varying components such as tropical SST and associated regions of convective activity dominate tropical – mid-latitude interactions." (lines 547-549).*

8) L604: Did authors wanted to say "it is still significant"?
*We have corrected this sentence following the anonymous reviewer's suggestion (line 607).*

9) L604: "stronger causal effect": stronger than what?
*We have clarified this sentence: "During La Niña summers, SAM exerts a stronger causal effect on the Tibetan High than over the entire 1979-2018 period, ..." (line 607-608).*

10) L641: "CEN provide an encoded predictive model" Please be more specific, it reads a bit strange. One could say it provides a statistical model but it feels that the authors somehow avoid using this term.
*Indeed CEN provides a statistical predictive model, we have added this term in the sentence (line 644).*

11) L644: "path coefficient β ~ 0.4, thus indicating potential for predictability." Doesn't it mean that in terms of the variance explained SAM explains about 16 % of the CGT variability? Thus, the statement is a bit too bold.
*We agree with the reviewer and we have modified the sentence, which now reads "Our analysis shows that at 4-weekly timescale, the effect of SAM on the CGT pattern has a path coefficient β ~ 0.4, thus indicating some potential for predictability." (lines 646-648).*

12) L645: Further or future studies instead of "Further work"
*Following the reviewer's suggestion, we have changed "further work" to "future studies" (line 645).*

13) Figure2 caption: timescaletimescale typo
*We thank the anonymous reviewer for his/her careful read. We have corrected these typos in the revised version of the paper.*